# In situ inorganic conductive network formation in high-voltage single-crystal Ni-rich cathodes

Xinming Fan[1], Xing Ou [1✉], Wengao Zhao [2,3✉], Yun Liu[1], Bao Zhang[1], Jiafeng Zhang[1], Lianfeng Zou[4], Lukas Seidl[2], Yangzhong Li[5], Guorong Hu[1], Corsin Battaglia [2] & Yong Yang [3✉]

High nickel content in $LiNi_xCo_yMn_zO_2$ (NCM, $x \geq 0.8$, $x + y + z = 1$) layered cathode material allows high specific energy density in lithium-ion batteries (LIBs). However, Ni-rich NCM cathodes suffer from performance degradation, mechanical and structural instability upon prolonged cell cycling. Although the use of single-crystal Ni-rich NCM can mitigate these drawbacks, the ion-diffusion in large single-crystal particles hamper its rate capability. Herein, we report a strategy to construct an in situ $Li_{1.4}Y_{0.4}Ti_{1.6}(PO_4)_3$ (LYTP) ion/electron conductive network which interconnects single-crystal $LiNi_{0.88}Co_{0.09}Mn_{0.03}O_2$ (SC-NCM88) particles. The LYTP network facilitates the lithium-ion transport between SC-NCM88 particles, mitigates mechanical instability and prevents detrimental crystalline phase transformation. When used in combination with a Li metal anode, the LYTP-containing SC-NCM88-based cathode enables a coin cell capacity of 130 mAh g$^{-1}$ after 500 cycles at 5 C rate in the 2.75-4.4 V range at 25 °C. Tests in Li-ion pouch cell configuration (i.e., graphite used as negative electrode active material) demonstrate capacity retention of 85% after 1000 cycles at 0.5 C in the 2.75-4.4 V range at 25 °C for the LYTP-containing SC-NCM88-based positive electrode.

[1] School of Metallurgy and Environment, Central South University, Changsha, P.R. China. [2] Empa, Swiss Federal Laboratories for Materials Science and Technology, Dübendorf, Switzerland. [3] School of Energy Research, Xiamen University, Xiamen, Fujian, P.R. China. [4] Environmental Molecular Sciences Laboratory, Pacific Northwest National Laboratory, Richland, WA, USA. [5] High Performance Computing Department, National Supercomputing Center in Shenzhen, Shenzhen, Guangdong, China. ✉email: ouxing@csu.edu.cn; wengao.zhao@empa.ch; yyang@xmu.edu.cn

Due to the large-scale deployment of electric vehicles and stationary battery systems, the global demand for Li-ion batteries (LIBs) is growing tremendously as they simultaneously satisfy the requirement of high energy and power density[1]. In particular, Ni-rich layered oxides $LiNi_xCo_yMn_zO_2$ (NCM, $x \geq 0.8$, $x + y + z = 1$) have attracted enormous attention as cathode materials for LIBs, owing to their reasonable cost and high practical energy density[2,3]. Following the successful commercialization of Ni-rich NCM with moderate Ni content ($x = 0.6/0.8$), the most effective approach to further increase the energy density and reduce production costs to further increase the Ni content while further lowering the Co content[4].

Currently, conventional NCM cathodes are composed of secondary microspheres consisting of aggregated primary NCM nanoparticles to obtain high tap and energy densities[5]. However, practical implementation of Ni-rich NCM with high Ni content is impeded by severe capacity fading and induced thermal issue during long-term cycling. Ni-rich layered NCM undergoes large anisotropic volume variations during cycling, compromising the mechanical stability and generating intergranular cracks[6,7]. Such intergranular cracks propagate along particle grain boundaries causing spalling off the secondary microspheres and their subsequent pulverization, eventually resulting in fast capacity fading[8]. Furthermore, as the electrolyte is facilitated to penetrate into the microspheres along intergranular cracks, the interphase area between the cathode and the electrolyte is significantly increased, which promotes electrolyte decomposition and the undesired phase transformation from the ordered layered NCM structure to the disordered rock-salt/spinel phase[9]. Intragranular crack formation becomes more pronounced as the Ni content exceeds 0.85 and is responsible for the continuous fading of the NCM cathode capacity[10]. Arranging and orienting the primary nanoparticles within the microsphere is a useful strategy to alleviate this issue[11]. For instance, quasi single-crystal layered NCM cathodes with primary particles of 2–5 μm diameter with a strong crystallographic texture were explored to minimize internal strain caused by the anisotropic volume variations during cycling[12,13], ultimately also decreasing electrolyte-induced corrosion due to the absence of grain boundaries and intragranular cracks[14]. Previously, we reported that micron-sized quasi single-crystal Ni-rich $LiNi_{0.83}Co_{0.11}Mn_{0.06}O_2$ particles can efficiently restrain the generation of intergranular cracks and alleviate the accumulation of parasitic interphase reactions, resulting in a significantly enhanced capacity retention of 84.8% after 600 cycles, while the poly-crystal NCM83 just maintains 57.4% applying the same measurement protocol[15].

Although the introduction of quasi single-crystal particles can enhance the cycling stability by suppressing the formation of micro/nanocracks, it is still challenging to achieve long-term stability when cycling to high cut-off voltages (>4.3 V vs Li/Li⁺) or temperatures higher than 25 °C as the rate of parasitic reactions at the cathode/electrolyte interphases increases. Moreover, it is difficult to achieve a desired rate capability due to the

prolonged lithium diffusion pathway within the thick bulk NCM microsized-particles[16,17]. Worse yet, in the highly delithiated state, the irreversible phase transition from the second hexagonal (H2) to the third hexagonal (H3) structure accompanies by a gradual loss of H3 phase, which causes an abrupt unit cell volume contraction and bulk structural collapse, even though the single-crystal architecture alleviates the formation of intragranular nanocracks during long-term cycling[18,19]. These cracks can not only release mechanical stress inside the primary particles, but also allow electrolyte penetration provoking the irreversible phase transition from H2 to H3, which aggravates capacity fading[20,21]. All of these unfavorable factors accumulate to serious performance deterioration of Ni-rich cathodes when cycled to high voltage or at high temperature, compensating the advantage of higher energy density.

Applying a surface coating with a high ionic conductivity is considered an effective strategy to stabilize the electrode/electrolyte interface[22–24]. In this work, we have designed an in situ modification strategy to construct a sodium-super-ion-conductor-type (NASICON-type) $Li_{1.4}Y_{0.4}Ti_{1.6}PO_4$ (LYTP) layer on the surface of single-crystal SC-NCM88 particles, forming a uniform and conformal 3D conductive network connecting the active particles. The LYTP modification framework helps to facilitate the lithium-ion (Li⁺) transport between NCM cathode particles as LYTP is a NASICON-type compound with one of the highest Li⁺ conductivities[25–27]. Compared to the radii of Li⁺/Ni²⁺ ions, the relatively larger radius of Y³⁺ tends to anchor on the surface of the cathode particles, avoiding Y³⁺ diffusion into the particle bulk, and thus plays a role in the 3D interconnected network rather than having a bulk doping effect on the bulk of SC-NCM88. Moreover, the trace Ti-doping at the surface/subsurface is helpful to suppress the disordered rock-salt/spinel phase formation due to the strong Ti-O bond, lowering the interfacial lattice mismatch and maintaining the surface structural stability[28]. Remarkably, LYTP-modified SC-NCM88 displays roughly 1.5 times higher Li⁺ conductivity than pristine SC-NCM88 due to the intrinsically fast Li⁺ transport in LYTP and the 3D ion-conducting network formed between the single-crystal particles, which is very effective in enhancing the reversible capacity and rate capability of SC-NCM88. Simultaneously, the electron conductivity of the LYTP-modified SC-NCM88 is also by a factor of 1.3 times compared to the pristine SC-NCM88 due to the interconnected LYTP network between the particles. Moreover, the conformal LYTP layer promotes high interface stability, as it is inert and prevents parasitic cathode/electrolyte reactions, maintaining the particles' structural integrity even under high voltage operation (≥4.4 V vs Li/Li⁺) and higher testing temperatures (55 °C) during long-term cycling. All these factors substantially boost the cycling stability and rate performance of single-crystal SC-NCM88 at high voltage operation. Therefore, the LYTP-modified SC-NCM88 delivers significantly improved cycling stability at 25 °C and 55 °C, and excellent rate capability up to 5 C (1 C = 200 mA g⁻¹). Noted, the pouch-type full cell with a practical areal capacity of 6.48 mAh cm⁻² achieves an excellent capacity retention of 85% after 1000 cycles when cycled to a high cut-off voltage of 4.4 V, maintaining a boosted discharge capacity of 170 mAh g⁻¹ and specific energy of 620 Wh kg⁻¹ on the active cathode material level.

## Results

**Synthesis and characterizations.** The in situ modification strategy to coat SC-NCM88 cathode particles with LYTP via a one-step calcination approach is illustrated in Fig. 1. As presented in Supplementary Fig. 1, the 1% LYTP@SC-NCM88 precursor consists of a tridimensional cluster sheets morphology with a

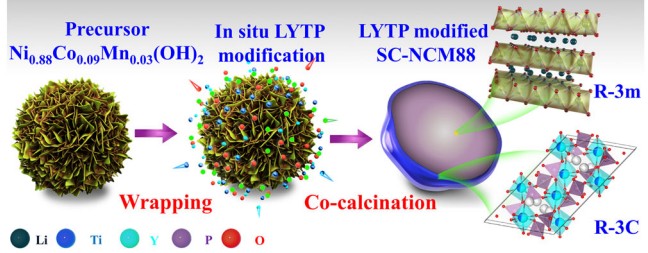

**Fig. 1 LYTP@SC-NCM88 synthesis process.** Schematic illustration of the synthesis method for LYTP-modified SC-NCM88 cathode.

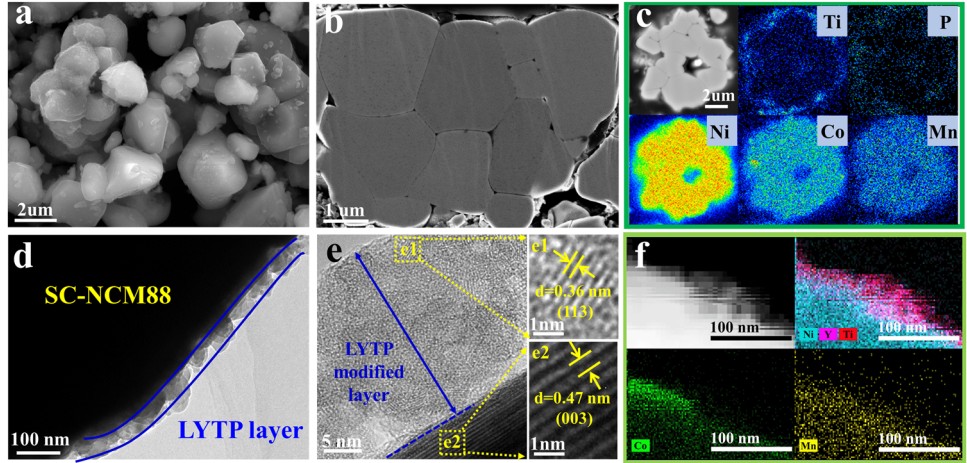

**Fig. 2 Representative electron microscopy images of LYTP@SC-NCM88. a** Overall and (**b**) cross-sectional morphologies derived from SEM images.
**c** Cross-section EPMA image of 1% LYTP@SC-NCM88 with the corresponding selected area LYTP mapping results of Ni, Co, Mn, Ti, and P elements.
**d** TEM, (**e**) HRTEM, and (**f**) STEM elemental mappings of Ni, Co, Mn, Y, and Ti for 1% LYTP@SC-NCM88.

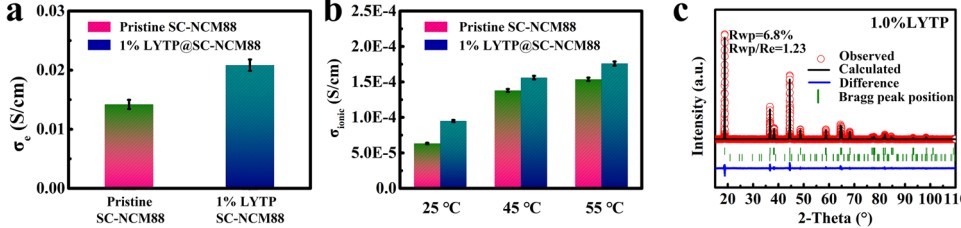

**Fig. 3 Conductivity measurements and structural characterization of LYTP@SC-NCM88.** The comparison of (**a**) electron conductivity and (**b**) Li-ion conductivity between pristine SC-NCM88 and 1% LYTP@SC-NCM88. Error bars represent the standard deviations of three measurements for conductivity. **c** The XRD Rietveld refinement of 1% LYTP@SC-NCM88.

secondary particle size of 3-5 μm. The LYTP precursors of Ti, Y, and P are uniformly distributed on the entire particle surface of $Ni_{0.88}Co_{0.09}Mn_{0.03}(OH)_2$. LYTP@SC-NCM88 precursor particles were transferred to a tube furnace for co-calcination to obtain LYTP-modified SC-NCM88 particles. After calcination at an optimized temperature of 820 °C, the precursors turn into micron-sized "single-crystal" particles with a crystalline LYTP modification (Fig. 2a and Supplementary Figs. 2–4). During calcination, particles tend to agglomerate, but subsequent crushing yields dense micron-sized individual single-crystal particles as shown in the cross-sectional SEM image (Fig. 2b). Energy-dispersive X-ray spectroscopy (EDS) mapping of the cross-sectional structure in Supplementary Fig. 5 shows that the Y and Ti anchor on the particles' surface. Additionally, cross-sectional electron-probe micro-analysis (EPMA) was applied to analyze the elemental distribution in the particles (Fig. 2c and Supplementary Fig. 6). It is clear that Ti, P, and Y mainly cover the particles surface, while Ni, Co, and Mn are found in the core of the particles, confirming the formation of quasi core–shell-structured LYTP@SC-NCM88 with a uniform LYTP layer. Moreover, gaps between particles are also filled with LYTP, demonstrating the formation of a 3D interconnected LYTP network, which is able to facilitate the $Li^+$-ion transport between the micron-sized SC-NCM particles.

Transmission electron microscopy (TEM) is employed to confirm the presence of the LYTP layer. A compact layer with a thickness of approximately 15 nm is homogeneously distributed on the surface of the pristine SC-NCM88 particle (Fig. 2d, e). In the magnified insets lattice fringes of 0.47 nm and 0.36 nm are

identified in the bulk and at the surface of the particles, corresponding to the (003) planes of SC-NCM88 and (113) planes of LYTP (Fig. 2e), respectively. EDS mapping in Fig. 2f further reveals the homogeneous distribution of Y and Ti, confirming the formation of a LYTP skin on the SC-NCM88 particle surface. Moreover, high-resolution XPS spectra of Y 3d, Ti 2p and P 2p further confirm that Y, Ti, and P exists in the 1% LYTP@SC-NCM88 (Supplementary Fig. 7). Besides, the elemental composition of the as-obtained powders was confirmed by inductively-coupled-plasma (ICP) analysis and is consistent with the nominal stoichiometry (Supplementary Table 1). Interestingly, according to a STEM-EDS line scanning result (Supplementary Fig. 8), Ti is not only found in the LYTP layer, but is also observed in the subsurface regions. Meanwhile, the scanning also clearly shows that the Ni, Co, and Mn contents barely change from the subsurface layer to the interior bulk. Thus, the LYTP modification results in a LYTP layer with a thickness of 5–20 nm covering the primary particles, but also leads to a Ti trace doping in the subsurface of the SC-NCM88 particles. Different from traditional coating methods, in situ modification of NCM precursor guarantees that the LYTP layer anchoring around the grain boundaries of SC-NCM88 during the co-calcination process, naturally constructing a 3D ion-conductive network by integrating the intrinsic fast $Li^+$ transport of LYTP.

The electronic conductivity of 1% LYTP@SC-NCM88 measured by the four-point probe method is higher than for the pristine SC-NCM88 powder, owing to the designed conductive framework (Fig. 3a). Benefitting from the high ionic conductivity of LYTP (Supplementary Fig. 9), the Li-ion conductivity of bulk

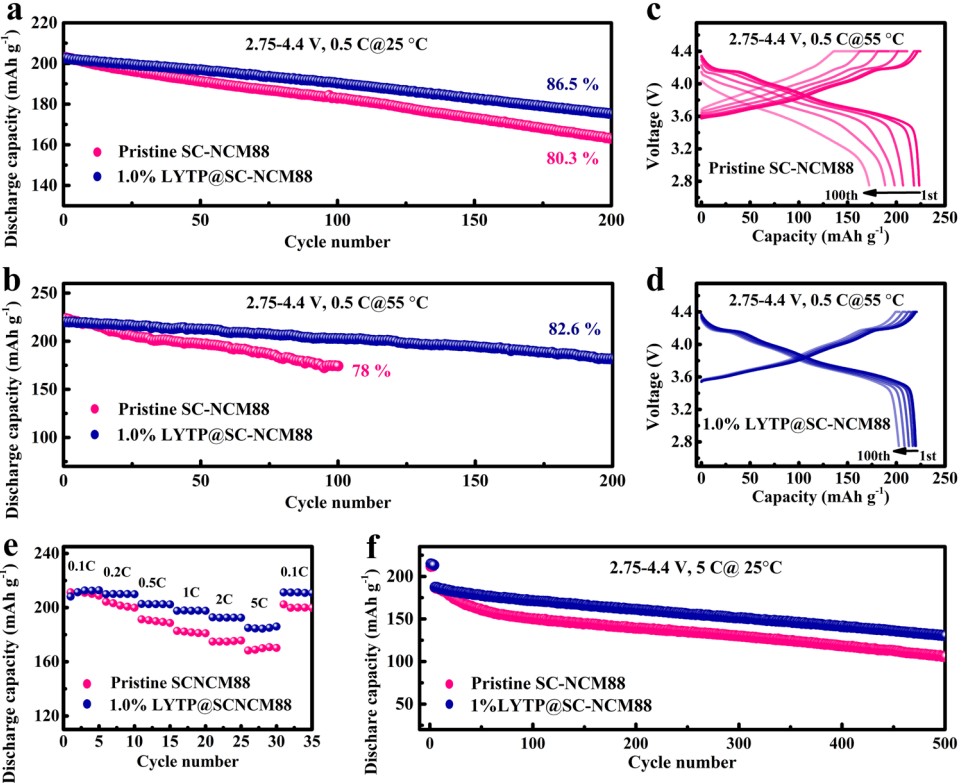

**Fig. 4 Electrochemical characterization of coin-type half cells.** Cycling stability of pristine SC-NCM88 and 1% LYTP@SC-NCM88 against a lithium metal anode at 0.5 C under testing temperature of (**a**) 25 °C and (**b**) 55 °C. Charge/discharge curves for (**c**) SC-NCM88 and (**d**) 1% LYTP@SC-NCM88 from 1st to 100th cycle at 55 °C. **e** Cycling capability at various current densities and (**f**) long-term cycling stability at 5 C for SC-NCM88 and 1% LYTP@SC-NCM88. All cells were cycled in the voltage range 2.75–4.4 V.

NCM88 single-crystal is increased due to the conformal LYTP modification for all measured temperatures from 25 °C to 55 °C, measured by using electrochemical impedance spectroscopy (EIS) with direct-current polarization (Fig. 3b)[29–31]. Numeric values of the electronic and ionic conductivity are summarized in Supplementary Table 2 and Table 3. The enhanced electronic and ionic conductivity for SC-NCM88 is playing an important role in achieving a higher reversible capacity and rate capability resulting from the 3D ion-conducting network between the individual particles.

The X-ray diffraction (XRD) patterns of pristine SC-NCM88, modified LYTP@SC-NCM88 and pure $Li_{1.4}Y_{0.4}Ti_{1.6}(PO_4)_3$ (LYTP) are displayed in Supplementary Fig. 10. Generally, the NASICON layer has a low crystallinity after calcination below 800 °C, but turns to the highly crystalline NASICON solid electrolyte above 800 °C. The resulting LYTP layer matches well with the NASICON-type reference pattern for $LiTi_2(PO_4)_3$ (JCPDS card No. 35-0754, R-3c space group). It is noted that the layered hexagonal structure of SC-NCM88 maintains unchanged within the appropriate amount of LYTP (≤1.5%), further indicating the successful fabrication of LYTP@SC-NCM88 cathode particles. However, the addition of 3% LYTP@SC-NCM88 results in the appearance of characteristic LYTP peaks at approximately 20.8° and 24.5°. The ordered layered structure of SC-NCM88 modified by the LYTP network is confirmed by the Rietveld refinement (Fig. 3c, Supplementary Fig. 11).

**Electrochemical performance**. To reveal the advantages of the LYTP modification, the cycling stability and rate capability of pristine SC-NCM88 and 1%LYTP@SC-NCM88 cathodes were cycled in half cells with a cut-off voltage of 4.4 V. As presented in Fig. 4a, the 1% LYTP@SC-NCM88 achieves a reversible capacity of 175 mAh g$^{-1}$ at 0.5 C after 200 cycles, corresponding to a capacity retention of 86.5% and coulombic efficiency of approximate 99.9% (Supplementary Fig. 12), while the pristine SC-NCM88 cathode displays only 163 mAh g$^{-1}$ after 200 cycles and 80.3% capacity retention under the same cycling condition. The 0.5% LYTP@SC-NCM88 and 3% LYTP@SC-NCM88 cathodes also exhibit rapid capacity fading and decreasing average discharge voltage compared to the 1% LYTP@SC-NCM88 (Supplementary Fig. 13a, b), emphasizing the 1% LYTP is the optimal amount as a coating skin. Consequently, the 1% LYTP@SC-NCM88 cathode delivers a much higher specific energy of 671 Wh kg$^{-1}$ at 0.5 C on cathode material level after 200 cycles than the pristine SC-NCM88 and the other modified cathodes (Supplementary Fig. 13c). The evolution of the corresponding charge/discharge curves is displayed in Supplementary Fig. 14, suggesting here a lower polarization for the 1% LYTP@SC-NCM88 cell. Furthermore, it is also noteworthy that the 1% LYTP@SC-NCM88 cathode is able to attain a capacity retention of 82.6% (182 mAh g$^{-1}$) with stable columbic efficiency (Supplementary Fig. 15) after 200 cycles even at high testing temperature of 55 °C (Fig. 4b), higher than the 78.0% (174 mAh g$^{-1}$) capacity retention of pristine SC-NCM88 cathode after 100 cycles. Comparison of the charge/discharge profiles of pristine SC-NCM88 (Fig. 4c) reveals the lowest polarization for 1% LYTP@SC-NCM88 (Fig. 4d), as the optimal thickness of the LYTP modification guarantees a high cathode/electrolyte interphase stability and maintain a benign Li$^+$ transport.

The rate capability was evaluated at various rates ranging from 0.1 C to 5 C (Fig. 4e). The 1% LYTP@SC-NCM88 delivers a higher reversible capacity comparable to the pristine SC-NCM88 at the same specific current. Even at a rate of 5 C, the 1%

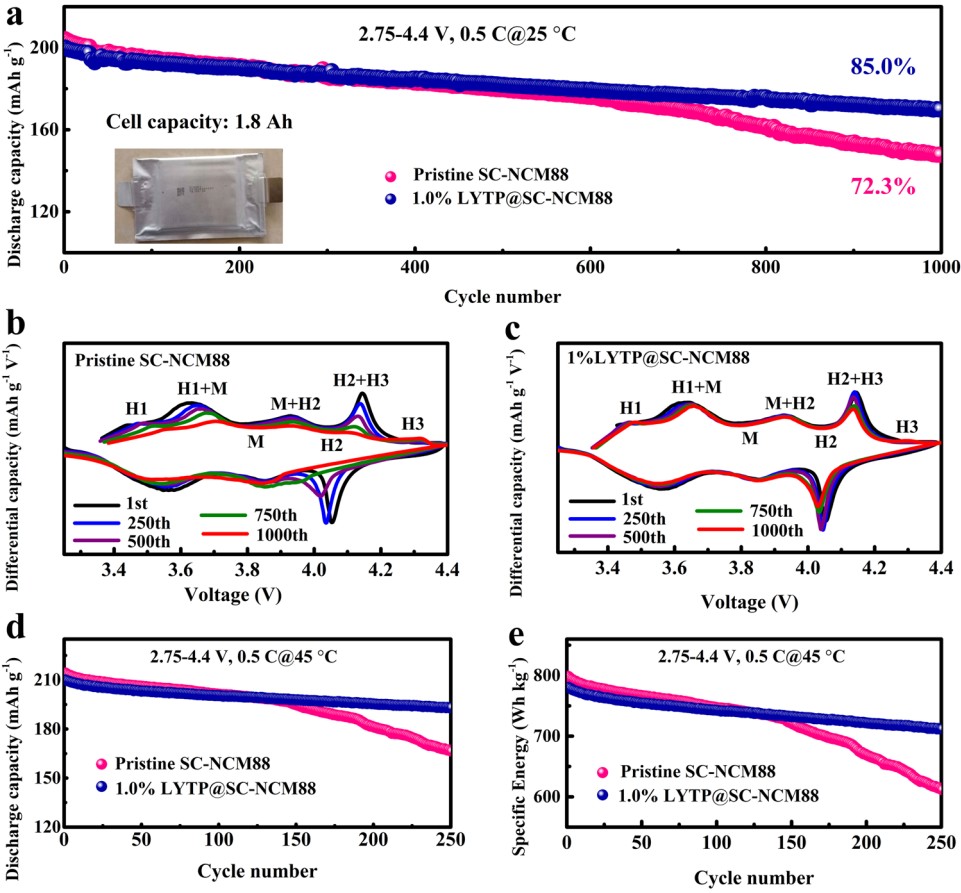

**Fig. 5 Electrochemical evaluation for pouch-type full cells. a** Cycling performances and (**b**, **c**) corresponding dQ/dV curves of the pristine SC-NCM88 and the 1% LYTP@SC-NCM88 against a graphite anode from the 1st cycle to the 1000th cycle. **d** Cycling performance and (**e**) specific energy for the pristine SC-NCM88 and the 1% LYTP@SC-NCM88 at an elevated temperature of 45 °C. All cells were cycled in the voltage range of 2.75–4.4 V.

LYTP@SC-NCM88 cathode delivers a reversible capacity of 177 mAh g$^{-1}$ (81.7% of the capacity at 0.1 C), higher than 164 mAh g$^{-1}$ of pristine SC-NCM88 (76.1% of the capacity at 0.1 C). Also during the long-term cycling at 5 C, the 1% LYTP@SC-NCM88 maintains a capacity of 130 mAh g$^{-1}$ after 500 cycles when cycled in the voltage range of 2.75–4.4 V, while the SC-NCM88 has a capacity of 106 mAh g$^{-1}$ applying the same measurement protocol (Fig. 4f). An improved cycling stability of the 1% LYTP@SC-NCM88 is also visible when comparing the columbic efficiency, average discharge voltage and specific energy as a function of cycle life (Supplementary Fig. 16). Obviously, the Li$^+$ transport channels provided by the 3D LYTP network in 1% LYTP@SC-NCM88 account for the increased rate performance and reversible capacity retention, even under the harsh voltage conditions. This suggests that the combination of single-crystal Ni-rich NCM with NASICON-type LYTP modification layer represents a promising strategy for high power applications in LIBs.

We also evaluated the cycling stability of 1% LYTP@SC-NCM88 in a pouch-type full cell against a graphite anode. The areal capacity of the double-side coated cathode is 6.48 mAh cm$^{-2}$ when coated on both sides and the total cell capacity amounts to 1.8 Ah, which is cycled in the voltage window of 2.75–4.4 at a rate of 0.5 C and 25 °C. Remarkably, the graphite/1% LYTP@SC-NCM88 cell maintains a discharge capacity of 170 mAh g$^{-1}$ after 1000 cycles with an enhanced capacity retention of 85.0% (Fig. 5a) and columbic efficiency of approximately 99.9% (Supplementary Fig. 17). Moreover, the average discharge voltage of the cell is above 3.64 V during the entire charge-discharge

process, and 82.1% of the initial specific energy of 620 Wh kg$^{-1}$ on the active cathode material level is maintained after 1000 cycles (Supplementary Fig. 18).

To reveal the electrochemical reaction reversibility and phase transition behavior during long-term charging/discharging, the corresponding dQ dV$^{-1}$ profiles were calculated by differentiating the 1st, 250th, 500th, 750th and 1000th charge/discharge curves. The pristine SC-NCM88 and 1% LYTP@SC-NCM88 cathodes all undergo a series of phase transformations from the hexagonal 1 (H1) phase to the hexagonal 3 (H3) phase via a monoclinic (M) and a hexagonal 2 (H2) phase (H1 → M → H2 → H3) during the delithiation, while the process happens reversibly during lithiation[32]. It is reported that the H2 → H3 phase transition results in an abrupt unit cell lattice contraction along the c-direction, thus the structure suffers from a severe mechanical strain with an anisotropic volume shrinkage at higher voltage[33]. Even though the single-crystal architecture effectively mitigates the formation of microcracks observed in the secondary particles, the electrochemical cycling can still generate nanocracks within the particles due to the crystal lattice contraction and expansion, eventually deteriorating the cathode material. However, compared to the pristine SC-NCM88 (Fig. 5b), a much higher reversibility across the H2 + H3 phase transition is observed for the LYTP-modified NMC88 (Fig. 5c), Explaining the significantly alleviates degradation due to the conformal LYTP modification. In order to distinguish the difference in the H2-H3 transition for the pristine SC-NCM88 and the 1% LYTP@SC-NCM88, the initial voltammetric cycles are overlaid in Supplementary Fig. 19. The peak intensity at ~4.15 V of 1% LYTP@SC-NCM88 is weaker than that of pristine SC-NCM88,

indicating the smaller anisotropic lattice volume variation after LYTP modification, which helps to reduce the irreversible H3 phase transformation. The corresponding charge/discharge curves of the 1% LYTP@SC-NCM88 cathode from the 1st to the 1000th cycle also display a higher reversible capacity and lower polarization (Supplementary Fig. 20), which may be ascribed to the improved $Li^+$ conductivity and stability of the LYTP protection layer.

A performance comparison of our pouch cell with other Ni-rich NCM cathodes reported previously (including commercial NCM cathodes) is shown in Supplementary Fig. 21 and Table 4[34–39]. The 1% LYTP@SC-NCM88 cathode delivers by far the highest reversible capacity while simultaneously offering a high stability when cycled at 0.5 C up to 4.4 V. This excellent result demonstrates the potential practical application of single-crystal cathode materials with ultrahigh Ni content. Also, when considering production costs, single-crystal Ni-rich cathodes materials are a serious contender to replace layered cathodes with higher cobalt content such as $LiCoO_2$. Pouch cells were also cycled at 45 °C to evaluate the cycling stability of SC-NCM88 and 1% LYTP@SC-NCM88 under accelerated aging conditions. In Fig. 5d, the SC-NCM88 and the 1% LYTP@SC-NCM88 exhibit a similar initial reversible capacity of approximately 210 mAh g$^{-1}$ at 0.5 C, approximately. After 250 cycles, the 1% LYTP@SC-NCM88 maintains a reversible capacity of 193 mAh g$^{-1}$ corresponding to a capacity retention of 91.9%. In contrast, the pristine SC-NCM88 retains a capacity of only 167 mAh g$^{-1}$ and inferior capacity retention of 77.8%. The corresponding discharge curves shown in Supplementary Fig. 22, suggest that more active $Li^+$ is preserved and less polarization is achieved in the 1% LYTP@SC-NCM88 cell than in the pristine SC-NCM88. Accordingly, the cell with 1% LYTP@SC-NCM88 delivers a promising specific energy of 712 Wh kg$^{-1}$ at the active cathode material level after 250 cycles, while the cell with the pristine SC-NMC88 cathode delivers 613 Wh kg$^{-1}$ (Fig. 5e). Thus, the significantly enhanced cycling stability at 45 °C is another benefit resulting from the LYTP modification, which stabilizes the cathode/electrolyte interface and ensures minimal capacity loss and maximum cycling life. Note that this is a pioneering report for single-crystal Ni-rich layered NCM full cells due to the enhanced cycling capability at high voltage operation.

**Operando XRD characterization and electron density of state analysis**. To understand how the LYTP modification improves the cycling capability and structural integrity, we investigated the structural evolution during the first two lithiation/delithiation cycles by employing operando XRD. Figure 6 shows operando XRD patterns of SC-NCM88 and 1% LYTP-SC-NCM88 recorded during the first cycle, while data for the first and the second cycles are presented in Supplementary Fig. 23. During the first cycle for the pristine SC-NCM88 cathode, the (003), (101), (006) and (107) peaks indicate good quality of the layered structure and display an obvious state-of-charge dependent shift (Fig. 6a). Specifically, the shift of the (003) reflection to lower angles for SC-NCM88 between 2.75 V and 4.1 V indicates a gradual expansion of the c-axis, corresponding to the H1-H2 phase transformation, which is induced by the columbic repulsion between the adjacent layers within the delithiated unit cell. At about 4.1 V, the (003) reflection splits into two peaks, suggesting H3 phase appearance coexisting with the H2 phase. Upon further delithiation, the H2-H3 phase transition induces a drastic contraction of the c-axis, leading to an aggressive shift of the (003) reflection to higher angles.

According to literatures for LiNiO$_2$-based layered cathodes[40,41], the interslab distance along the c-axis significantly shrinks in the highly delithiated state. This is confirmed by the abrupt shift of the (003) reflection indicating a shrinkage of the c

lattice parameter. In addition to that, the continuous removal of $Li^+$-ion between the transition-metal layers weakens the pillaring effect, promoting the structure collapse and an abrupt variation in the interslab distance. In contrast to that, the 1% LYTP@SC-NCM88 demonstrates a pseudo-single-phase H2-H3 transition and maintains a relatively more stable structure at high voltage (Fig. 6b). Compared to the pristine SC-NCM88 with the maximum angle shift of 1.229° during this pseudo-single-phase H2-H3 transition (Fig. 6c), the (003) reflection of 1% LYTP@SC-NCM88 only shifts by a lower maximum angle of 0.937° (Fig. 6d), illustrating that the contraction along the c-axis of 1% LYTP@SC-NCM88 is reduced by 1.5%. The evolution of the c lattice parameter for pristine SC-NCM88 and 1% LYTP@SC-NCM88 during delithiation is summarized in Fig. 6e, confirming that the LYTP modification reduces the contraction along the c-axis. Therefore, the pronounced c-axis contraction observed for the pristine SC-NCM88 results in severe structure deterioration at high voltage, due to the irreversible H2-H3 phase transition[42]. In contrast, for the 1% LYTP@SC-NCM88, although the H2-H3 transition can still be observed, with smaller differences in the shape and position of the (003) reflection, which may be ascribed to the suppression of lattice mismatch between reconstructed surface and interior bulk under high voltage[43,44].

Moreover, compared with the deviation from its original 2θ position after full discharge for pristine SC-NCM88, the 1% LYTP@SC-NCM88 cathode displays a more symmetric evolution with respect to cell voltage during the first charge/discharge cycle, confirmed by the diffraction peaks returning back to their original position. During the subsequent second cycle, the XRD patterns exhibit similar phase transition with high reversibility (Supplementary Fig. 23). These findings illustrate that the LYTP modification can suppress the H2-H3 transformation and improve the reversibility of the phase transition, which reduces the electrode polarization and improves the structural integrity during the delithiation/lithiation process.

To get insights into the electronic properties of the 3D conductive LYTP network, density functional theory (DFT) calculation was performed to investigate the surface transformation of SC-NCM88 before and after modification with LYTP[28,45]. The total and partial density of state (DOS) for each element (Supplementary Figs. 24, 25) are also calculated by the DFT method. Normally, the electrical conductivity is represented by the electron density of states near the Fermi level (Fig. 7a)[46]. After LYTP modification of the SC-NCM88, the introduced 2p state of Ti, 3d state of Y, and 2p state of P atoms significantly increase the electron density of states near the Fermi level (Fig. 7b), which induces the left shift of overall DOS and increases the electron concentration, indicating the enhancement of its electronic conductivity. Therefore, it is concluded that the 3D conductive LYTP modification network enhances the electronic conductivity of 1% LYTP@SC-NCM88.

Furthermore, the charge differences for four types (pristine NCM, surface Ti-atoms doping, LYTP nanolayer coating, NCM with surface doping and LYTP coating) are also calculated and displayed in Fig. 7c and Supplementary Fig. 26, respectively. In comparison, there is an obvious electron transfer phenomenon between Ni, Co, Ti and the surrounding O atoms, which becomes more obvious with the introduction of surface Ti-doping and LYTP coating (Fig. 4d, e). Additionally, the degree of electron accumulation for Ti-O is higher than that of Ni–O and Co–O, which indicates that Ti-O has a stronger chemical bond, thus stabilizing the lattice oxygen during lithiation/delithiation. Based on the difference charge density of atomic configuration, the Bader charge transfer of the four types is also calculated, as displayed in Fig. 7f, g and in Supplementary Fig. 27. Through the Bader charge transfer analysis, it is also seen that Ti atom

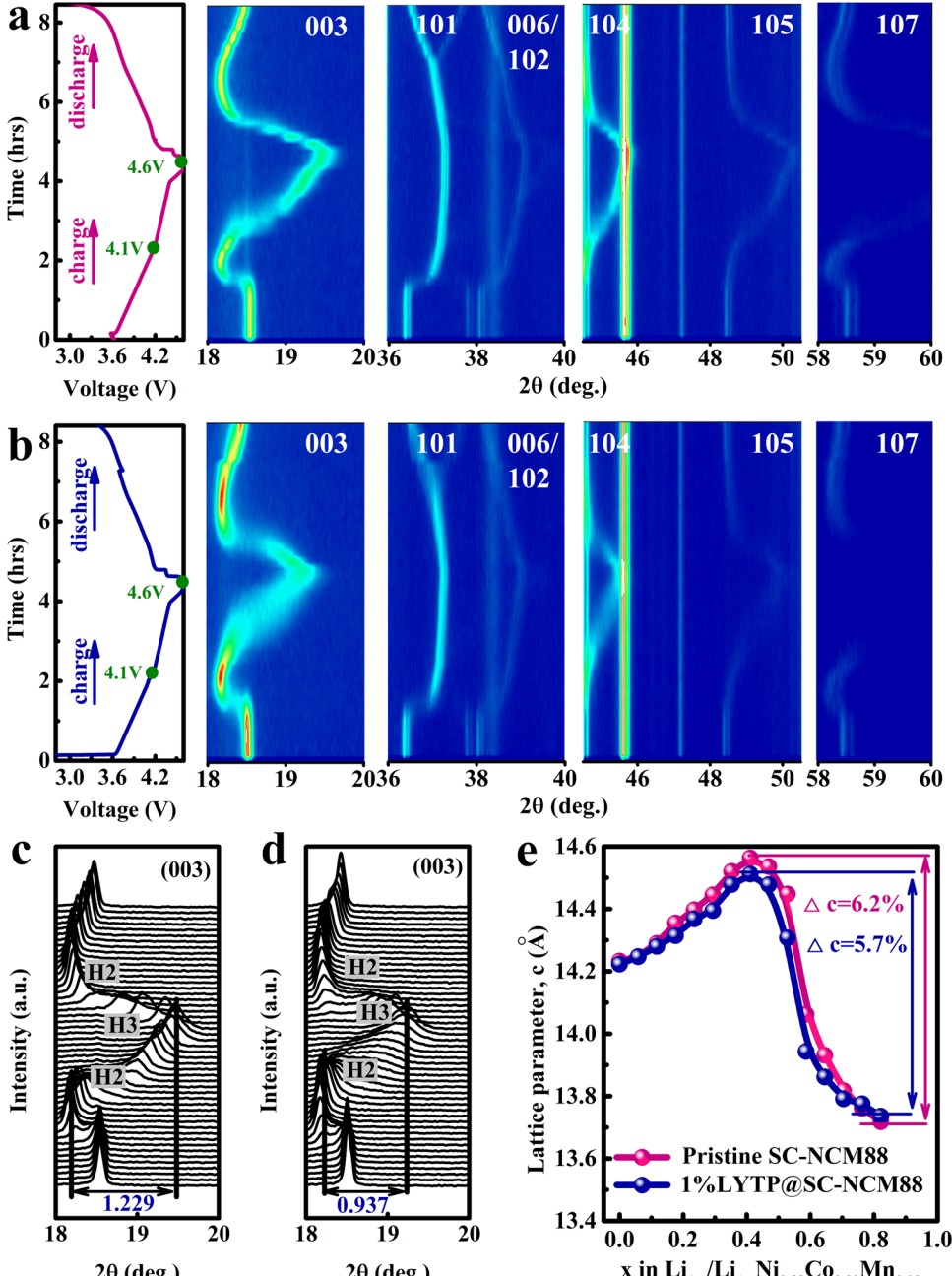

**Fig. 6 Investigation of phase transitions during cycling.** Operando XRD characterization of the full contour plots and selected line patterns for (**a**, **c**) SC-NCM88 and (**b**, **d**) 1% LYTP@SC-NCM88 cathodes during the initial cycle in the voltage range of 2.75–4.6 V. **e** The variation of the c-axis parameter during charging for pristine SC-NCM88 and 1% LYTP@SC-NCM88.

transfers more electrons than Ni and Co atoms, which can enhance the surface oxygen stability. This result is consistent with the previous differential charge density results. During the long-term cycling in a temperature range of 25 °C to 55 °C and high cut-off voltage, the improved surface stability will mitigate the lattice oxygen release. Therefore, it can suppress the unwanted transformations from layered structure to spinel/rock-salt phases, which improves the electrochemical and thermal performance.

**Ex situ post-mortem electrode measurements.** To get a deeper understanding of the mechanism leading to the enhanced cycling stability of pouch cells with LYTP-modified SC-NCM88, the cells were disassembled after 200 cycles in 2.75–4.4 V for a post-

mortem analysis. Firstly, A time-of-flight secondary ion mass spectrometry (TOF-SIMS) was applied to analyze the surface of cycled cathodes[47,48]. Depth profiling of the cathode–electrolyte interface (CEI) layer was performed by tracking the organic species ($C_2HO^-$, $POF_2^-$, $C_2F^-$ and $PO_3^-$) variation at the surface, which are generated by electrolyte salt and solvent decomposition during cycling (Fig. 8a–d). Transition-metal fluorides ($NiF_3^-$, $CoF_3^-$, $MnF_3^-$ and $^6LiF_2^-$) distribute uniformly from surface to subsurface of the cathode (Fig. 8e–h), which are generated by cathode dissolution due to attack by acidic species (e.g. HF) and due to the undesired phase transformation from layered to disordered rock-salt structure[49]. As can be seen from Fig. 8a–h, all these fragments exhibit a lower concentration for the 1% LYTP@SC-NCM88 cathode as compared to the unmodified SC-

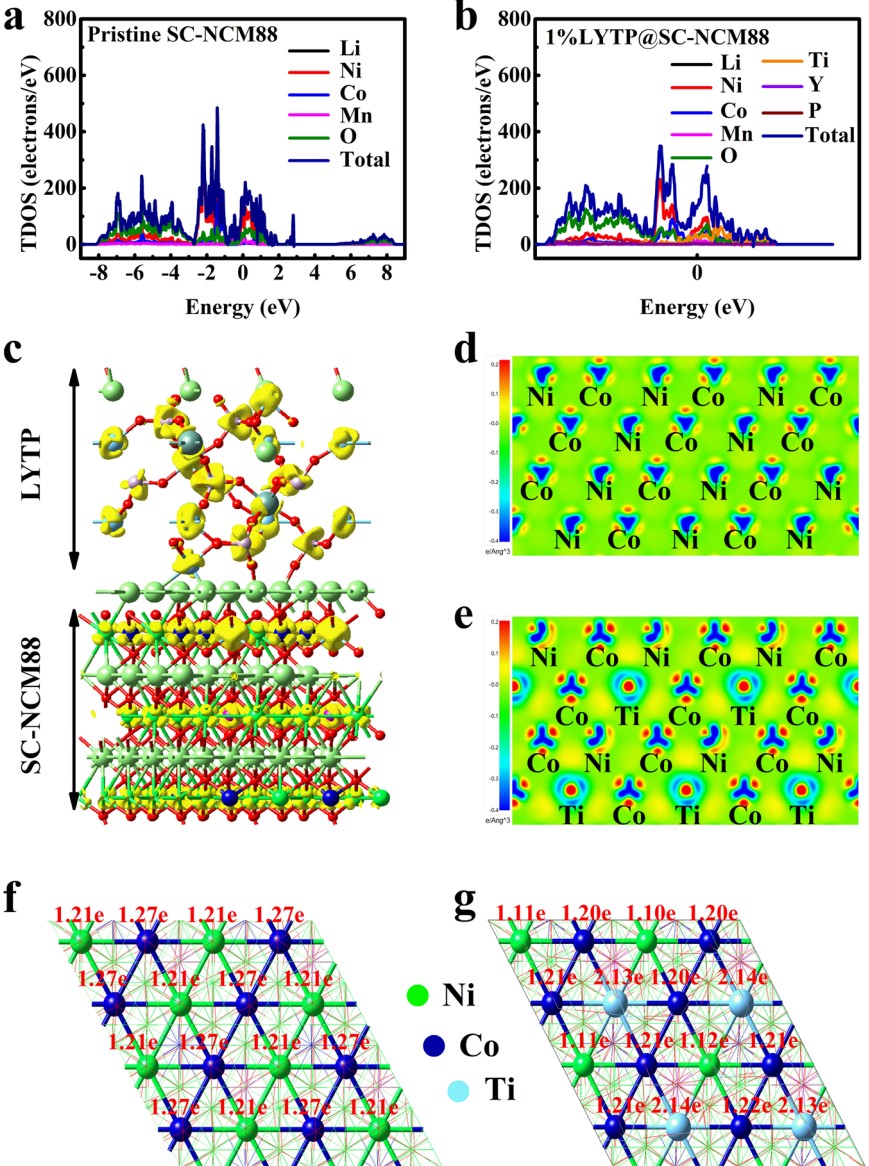

**Fig. 7 Density functional theory calculation.** The total and partial density of states plots for (**a**) pristine SC-NCM88 and (**b**) 1% LYTP@SC-NCM88. **c** Relaxed crystal structures of 1% LYTP@SC-NCM88. The corresponding 2D charge difference scheme and Bader charge transfer for (**d**, **f**) pristine SC-NCM88 and (**e**, **g**) 1% LYTP@SC-NCM88.

NCM88 cathode, suggesting suppressed cathode/electrolyte interphase parasitic reactions.

Additionally, the near-surface chemical composition of the cycled cathodes and anodes were examined by XPS[50,51]. Characteristic peaks corresponding to C−C, C−H, C−O, C=O, and OCO$_2$ bonds are observed in the C 1$s$ spectra for both pristine SC-NCM88 and 1% LYTP@SC-NCM88 cathodes (Fig. 8i)[52]. Compared to the pristine SC-NCM88, weaker peak intensities are observed for the peaks associated with C−O, C=O, and OCO$_2$ of the 1% LYTP@SC-NCM88 cathode, suggesting a lower electrolyte decomposition amount on the surface of the LYTP modification. Moreover, the O 1$s$ peaks associated with Li$_2$CO$_3$ and ROCO$_2$Li (Fig. 8j) are detected for the pristine SC-NCM88 cathode with stronger intensity, which is a convincing evidence for the higher carbonate decomposition accumulation on the raw cathode surface[49]. More evidence from fragments are in good agreement with F 1$s$ (Fig. 8k) and P 2$p$ spectra (Fig. 8l), such as LiF and Li$_x$PO$_y$F$_z$ species.

The characteristic species derive from the electrode/electrolyte side reactions and are found to consist of the passivating solid electrolyte interphase (SEI) film of graphite. The XPS spectra of cycled graphite anodes further confirm lower electrolyte decomposition on the 1% LYTP@SC-NCM88 surface compared to SC-NCM88 cathode. A stronger intensity in the O 1$s$ spectra associated with Li$_2$CO$_3$ and ROCO$_2$Li is observed on the cycled graphite against the pristine SC-NCM88 cathode (Supplementary Fig. 28a), which exhibits similar phenomenon for both peaks of Li$_x$PO$_y$F$_z$ and LiF in F 1$s$ spectrum (Supplementary Fig. 28b) and Li$_x$PO$_y$F$_z$ peak in P 2$p$ spectrum (Supplementary Fig. 28c). On the contrary, the cycled graphite paired with 1% LYTP@SC-NCM88 displays small amounts of Li-containing components. Thus, it is safely concluded that the LYTP modification efficiently mitigates electrolyte decomposition during high voltage operation. This conclusion is further supported by EIS tests (Supplementary Fig. 29)[53,54]. The charge transfer resistance (R$_{ct}$) of the pristine SC-NCM88 cell increases significantly after 200 cycles, indicating a sluggish ion transport through the cathode–electrolyte interface.

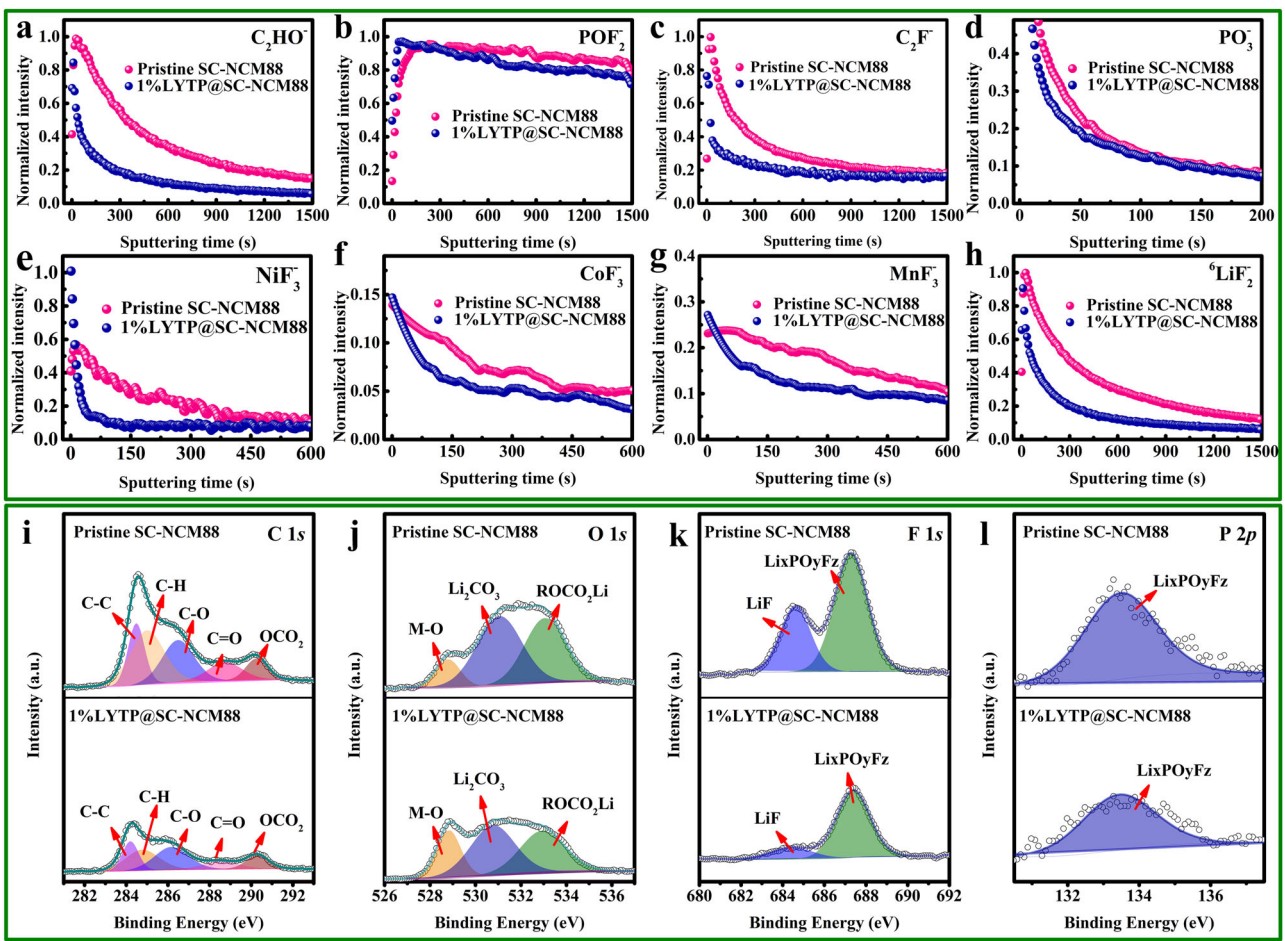

**Fig. 8 Ex situ surface chemistry characterizations of cycled pouch cell positive electrodes.** TOF-SIMS depth profiles of the near-surface chemical composition for (**a**) $C_2HO^-$, (**b**) $POF_2^-$, (**c**) $C_2F^-$, (**d**) $PO_3^-$, (**e**) $NiF_3^-$, (**f**) $CoF_3^-$, (**g**) $MnF_3^-$ and (**h**) $^6LiF_2^-$. XPS spectra of (**i**) C 1s, (**j**) O 1s, (**k**) F 1s and (**l**) P 2p elements for the pristine SC-NCM88 and 1% LYTP@SC-NCM88 cathodes after 200 cycles from 2.7 V to 4.4 V.

However, the increase of $R_{ct}$ of the 1% LYTP@SC-NCM88 cell demonstrates only a mild variation without sharp increase, due to the stable electrode-electrolyte interface with robust charge-transfer kinetics after LYTP modification (Supplementary Table 5).

To further elucidate the boosted cycling performance of 1% LYTP@SC-NCM88, the structural changes of the cycled cathodes are investigated, especially for the electrode/electrolyte interface. The cycled electrodes are extracted from the pouch-type full cell after 200 cycles in the voltage of 2.75–4.4 V at 0.5 C (vs. graphite). For the pristine SC-NCM88, its layered structure is disrupted and is completely transformed into the rock-salt phase at the outmost particle surface with a thickness that exceeds 15 nm (Fig. 9a). In contrast, it is clear that the LYTP modification with a thickness of 17.5 nm is still conformally anchored on the surface of SC-NCM88 particles (Fig. 9b and Supplementary Fig. 30), although 1% LYTP@SC-NCM88 sample exhibits a rough and fuzzy surface. The interplanar spacings at surface and interior bulk are indexed to (113) plane of LYTP and (003) plane of SC-NCM88, respectively, suggesting that the layered phase is effectively retained without obvious structural degradation. Moreover, the intact LYTP modification layer is also confirmed by STEM elemental mappings in Supplementary Fig. 31. The representative Ti element uniformly anchors on the surface of SC-NCM88, demonstrating that the LYTP modification layer maintains its chemical stability and structural integrity during long-term cycling.

Furthermore, inspection of the cross-sectional SEM image reveals numerous nanocracks inside the unmodified SC-NCM88 particles (Fig. 9c). Moreover, the disordered rock-salt phase $Fm\bar{3}m$ is detected at the vicinity of nanocracks (Supplementary Fig. 32), caused by parasitic cathode/electrolyte interphase reactions. Besides, the evolution of the intraparticle structure is investigated using (scanning) transmission electron microscopy ((S)TEM) on cross-sectional particles prepared by a focused ion beam (FIB). For comparison, two regions from the near surface to interior bulk in the range of 100 nm (Fig. 9d) are selected. Only $Fm\bar{3}m$ rock-salt phase is observed at near surface of region 1 (Fig. 9e), as confirmed by high-angle annular dark-field (HAADF) images. Even extending into the interior, the disordered spinel phase still occupies region 2 (Fig. 9f), indicating the severe irreversible phase transition of the pristine SC-NCM88. This phenomenon is typically observed in the polycrystalline Ni-rich layered oxides[55,56], indicating that the parasitic interface reactions also occur on single-crystal Ni-rich NCM at high voltage operation.

However, the 1 % LYTP@SC-NCM88 particles maintain their dense bulk structure without visible cracks (Fig. 9g). Again, two representative regions are selected to investigate the phase transition at the surface of the 1 % LYTP@SC-NCM88 particles (Fig. 9h). It is noted that 1% LYTP@SC-NCM88 particles present a standard layered structure extending from the bulk propagating to the outermost surface. Although the disorder layered phase is detected in region 3 (Fig. 9i), the ordered layered structure is well-maintained without any structural damage in region 4 (Fig. 9j).

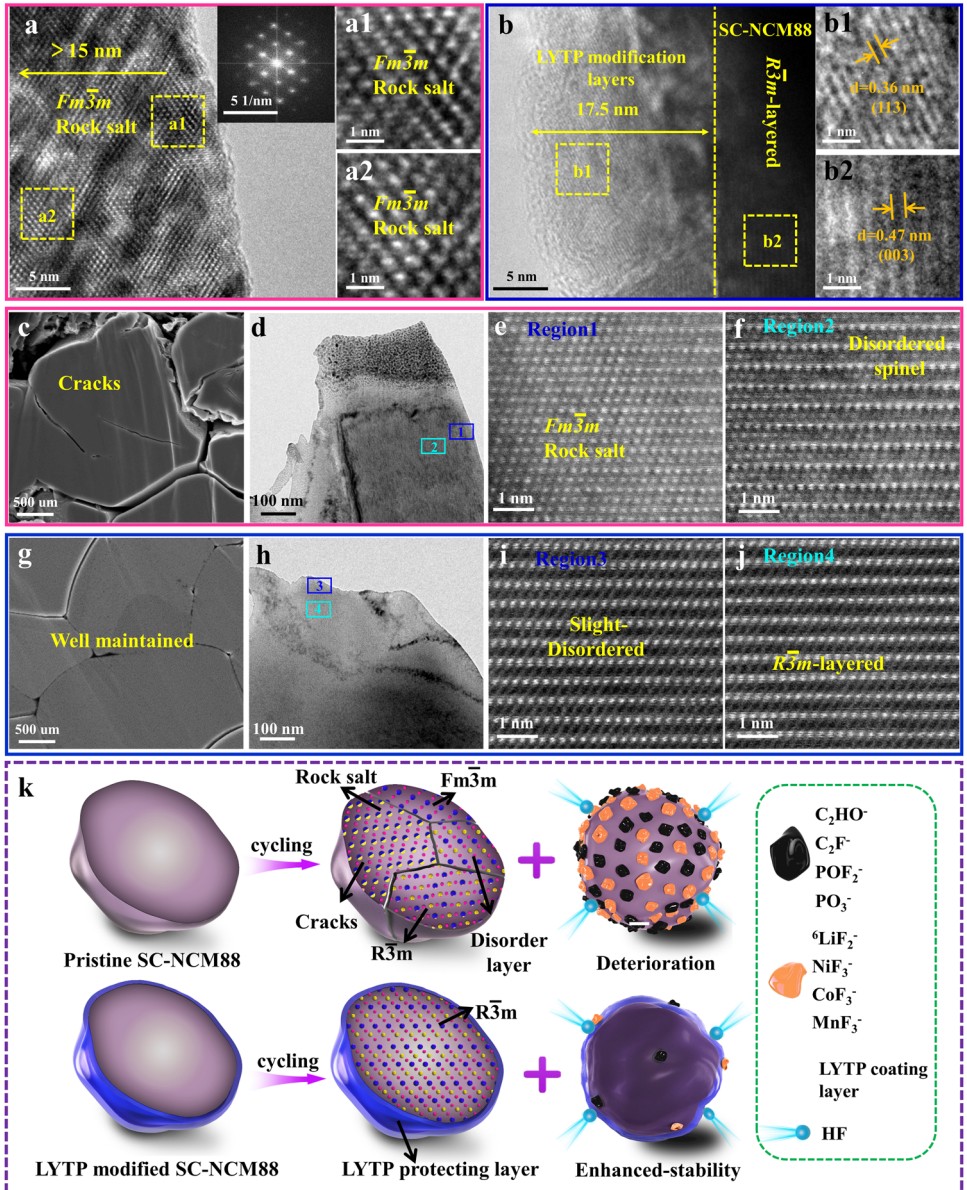

**Fig. 9 Intraparticle structural evolution after pouch cell long-term cycling.** Post-mortem HRTEM and magnified HRTEM at selected area images for (**a**, **a1**, **a2**) pristine SC-NCM88 and (**b**, **b1**, **b2**) 1% LYTP@SC-NCM88 after 200 cycles. Cross-sectional SEM images of (**c**) pristine SC-NCM88 and (**g**) 1% LYTP@SC-NCM88. Low-magnification HAADF-STEM image of FIB-cross-section for the surface region and magnified HAADF-STEM images taken from the corresponding surface areas for (**d**–**f**) pristine SC-NCM88 and (**h**–**j**) 1%LYTP@SC-NCM88 after 200 cycles. **k** Schematic illustration of the structure evolution and internal crack difference for pristine SC-NCM88 and 1% LYTP@SC-NCM88 particles during prolonged cycling.

This distinction illustrates that the LYTP modification layer enable to mitigates the disordered spinel/rock-salt phase formation, achieving an improved structural stability and enhanced electrochemical performance even under the harsh conditions.

The proposed structural evolution of the pristine SC-NCM88 and the 1% LYTP@SC-NCM88 particles during electrochemical cycling is schematically summarized in Fig. 9k. During long-term cycling, both samples subject to c-lattice parameter contraction and expansion caused by lithium extraction/insertion. The direct contact of the SC-NCM88 particle with the electrolytes leads to severe transition-metal dissolution and side reactions. These processes easily trigger the surface phase transformation from ordered layered to disordered spinel/rock-salt structure, which is in turn responsible for the formation of intragranular cracks, consequently leading to the further irreversible phase transition near these cracks with mismatch lattice. These issues are significantly suppressed by the protective LYTP modification, which prevents direct contact of SC-NCM88 particle with the electrolyte, effectively suppressing the transition-metal dissolution and formation of intragranular cracks. Furthermore, the LYTP layer is able to mitigate the interfacial lattice mismatch and the unavoidable H2-H3 phase transition due to strong Ti-O bond on the surface/subsurface of SC-NCM88, substantially improving the reversibility of the phase transitions and the structural integrity during the delithiation/lithiation process. More importantly, as a highly conductive NASICON-type ion conductor, LYTP layers with intimate contact and uniform distribution facilitate the construction of a 3D conductive network, not only providing a rapid ion diffusion pathway between adjacent NCM particles with reduced electrode polarization, but also inducing the unobstructed $Li^+$-ions transport from the outermost surface extending to inner bulk.

## Discussion

In summary, we demonstrated an in situ modification strategy to significantly improve the rate capability and cycling stability of a Ni-rich SC-NCM88 cathode by integrating a NASICON-type 1% LYTP interparticle network. Particularly, the 1% LYTP@SC-NCM88 maintains an enhanced cycling stability in temperature from 25 °C to 55 °C and high cycling rate in both coin-type half cells and pouch-type full cells. It is noteworthy that the pouch-type full cell delivers a remarkable capacity retention of 85% after 1000 cycles at 0.5 C charged up to 4.4 V vs a graphite anode with an enhanced discharge capacity of 170 mAh g$^{-1}$ and specific energy density of 620 Wh kg$^{-1}$ at the active material level. The remarkable cycling capability as well as the significantly improved interphase stability and intrinsic structure under harsh cycling conditions are attributed to the following reasons. (1) The protective LYTP coating and the simultaneous Ti trace doping exert a synergistic effect suppressing the disordered spinel/rock-salt phase formation and lattice mismatch between well-ordered layered and disordered structure, which substantially alleviates the c-lattice parameter contraction, and improving the reversibility of the H2-H3 phase transition. (2) The undesired intra-granular/intergranular cracking is obviously alleviated due to the reduced c-axis contraction and robust mechanical properties of conformal LYTP modification. (3) The LYTP modification layer facilitates the Li$^+$ transportation and enables a highly reversible capacity due to the high Li$^+$ conductivity of the 3D network between cathode particles. The key findings provide a guideline for developing high-energy-density single-crystal Ni-rich NCM cathodes operated under harsh conditions (≥ 4.4 V vs Li/Li$^+$ and at 45 °C) and at practical areal capacities (>3 mAh cm$^{-2}$) through building a robust 3D network with high Li$^+$ conductivity between the NCM particles.

## Methods

**Material synthesis**. The Ni$_{0.88}$Co$_{0.09}$Mn$_{0.03}$(OH)$_2$ precursor was synthesized via using a hydroxide co-precipitation method. The mixed solution (2 mol L$^{-1}$) of NiSO$_4$·6H$_2$O, CoSO$_4$·7H$_2$O and MnSO$_4$·5H$_2$O with a molar ratio of Ni: Co: Mn = 88: 9: 3 was pumped into a continuously stirred tank reactor (50 L) under inertial N$_2$ atmosphere. Next, the precipitation agent NaOH solution (5 mol L$^{-1}$) and chelating agent NH$_3$·H$_2$O (4 mol L$^{-1}$) were fed into the tank reactor separately. It is noted that the pH value (11.4), the reaction temperature (50 °C), and the stirring speed (500 rpm) were precisely controlled and maintained constantly. For the preparation of the LYTP-modified Ni$_{0.88}$Co$_{0.09}$Mn$_{0.03}$(OH)$_2$, 0.47 g LiNO$_3$, 0.75 g Y(NO$_3$)$_3$· 6 H$_2$O, and 1.44 g H$_3$PO$_4$ (the molar ration of Li: Y: P = 1.4: 0.4: 3) and a stoichiometric ratio of 2.68 g Ti(C$_4$H$_9$O)$_4$ were dissolved and mixed homogeneously in ethyl alcohol solutions. After that, the resultant spherical Ni$_{0.88}$Co$_{0.09}$Mn$_{0.03}$(OH)$_2$ (187.9 g) was added into the aforementioned solution with mild stirring. To get LYTP@SC-NCM88, this LYTP-precursor@Ni$_{0.88}$Co$_{0.09}$Mn$_{0.03}$(OH)$_2$ was thoroughly mixed with 90 g LiOH·H$_2$O (Li: M ratio = 1.06:1) and calcined at 820 °C for 10 h in oxygen atmosphere. The obtained material was finely ground by an air-jet mill and was stored in a vacuum package. After calcination, the LYTP mole fraction in the LYTP@SC-NCM88 composite was determined according to the rechecked ICP results (Supplementary Table 1). For 1.0%LYTP@SC-NCM88 sample, the mole ratio of LYTP: SC-NCM88 is 0.007:1, which is transformed into the mass ratio of 0.01:1. The SC-NCM88 modified with different LYTP contents (abbreviated as "x% LYTP@SC-NCM88") can be achieved by regulating the mole ratio of SC-NCM88 and LYTP. For comparison, pristine SC-NCM88 powder was obtained through the same annealing process without the addition of LYTP.

**Material characterizations**. Structure information was detected by X-ray diffraction (Rint-2000, Rigaku type X-ray diffractometer), which was analyzed by the Rietveld refinement program (General Structure Analysis System (GASA) software). For the operando XRD measurement, a home-made Swagelok cell coupled with a Be window was used for X-ray-transparency. The morphology and microstructure were investigated by scanning electron microscopy (SEM, JSM 6400, JEOL), transmission electron microscopy (TEM, JEOL 2100 F, JEOL), and spherical aberration corrected transmission electron microscopy (ACTEM, FEI Titan G2 80-200 ChemiSTEM). Before TEM and ACTEM measurement, samples were treated by focused ion beam (FIB, SCIOS, FEI). Chemical compositions were conducted by X-ray photoelectron spectroscopy (XPS, Thermo Fisher ESCALAB 250Xi), time-of-flight secondary ion mass spectroscopy (TOF-SIMS, ION-TOF), and inductively-coupled-plasma (ICP, OPIMA 8300, Perkin Elmer).

**Electrochemical measurements**. The cathode electrodes were prepared from a mixture of active material (94 wt%), carbon black (2 wt%), KS-6 (2 %) and PVDF (2 wt%), dissolved in N-methyl-1,2-pyrrolidone solvent (NMP) to obtain the homogeneous slurry. The thickness of positive electrodes for coin cell and pouch cell were about 43–44 μm and 107–108 μm after calendaring, respectively. The cells were assembled in an argon-filled glovebox (the contents of both O$_2$ and H$_2$O were ≤0.1 ppm) with 1 M LiPF$_6$ in ethyl carbonate/diethylene carbonate (EC/DEC, 1:1 in volume) and a Celgard 2400 as the electrolyte and separator, respectively. For 2032-type coin cells, Li metal was applied as the counterpart electrode, while the mass loading of cathode material was approximately 8.5 ± 0.15 mg cm$^{-2}$. 120 mg of electrolyte solution are used for a single coin cell assembly. For pouch-type full cells, graphite (BTR New Energy Material Ltd) was used as counterpart anode. The height, width and length of the pouch cell is 4 mm, 60 mm and 120 mm, respectively. The cell design capacity is about 1800 mAh, thus the mass loading weights of cathode and anode were about 32.4 mg cm$^{-2}$ and 21.0 mg cm$^{-2}$ on both sides with designed capacity of 200 and 340 mAh g$^{-1}$, respectively. A porous polymer (UBE, UP3074) and 1.1 M LiPF6 EC/EMC/DEC (3:5:2 vol%) + 1 wt%VC as electrolyte were used in the pouch cell. The pouch cells were assembled in a winding process. The capacity balance of anode to cathode was approximately 1.14. Each pouch cell contained 10 layers of anode (73.5 mm × 565 mm in size) and 9 layers cathode (68 mm × 445 mm in size). The average electrolyte injection coefficient and electrolyte retention coefficient for pouch cell is 4.0 g Ah$^{-1}$ and 3.9 g Ah$^{-1}$, respectively. Galvanostatic charge/discharge measurements were tested within a voltage of 2.75–4.4 V at various temperatures (from 25 °C to 55 °C). The relative humidity for cycle testing is less than 65 RH%. The specific capacity is calculated based on the SC-NCM88 active materials for both coin-type cells and pouch-type cells. The discharge average voltage can be obtained from the battery tester, which is calculated by dividing the specific energy by the specific capacity. The specific energy data of material level could be directly got from the Landt/Neware battery tester, which is calculated by integrating the area under the related voltage-discharge capacity profile. The electrochemical tests are carried out in the environmental chamber and the error of the temperature is ± 2 °C. Cyclic voltammetry was performed with a sweep rate of 0.1 mV s$^{-1}$ and within 2.75 V to 4.6 V on a CHI660E electrochemical workstation. For the electrochemical impedance spectroscopy test, a frequency range of 100,000 Hz to 0.01 Hz (10 data points per decade of frequency) at a 5 mV amplitude was applied on a Bio-Logic EC-LAB SP-300 electrochemical workstation.

**Density functional theory calculations**. Density functional theory (DFT) calculation was conducted by applying the Vienna Ab-initio Simulation Package[57], while the electron-ion interaction, exchange and correlation functionals were described by projector augmented-wave (PAW) pseudopotentials, Perdew–Burke–Ernzerhof (PBE) version of the generalized gradient approximation (GGA), respectively[58]. In addition, the DFT + U method was also introduced to describe the electronic properties and defect states. The value of U given to Mn, Co, and Ni ions were 4.0 eV, 3.5 eV, and 5.77 eV, respectively[59]. In the DFT calculation, the energy cut-off of 500 eV was adopted for wave functions expansion. For the Brillouin zone integration, a 3 × 3 × 1 k-grid mesh and a 5 × 5 × 3 k-grid mesh for geometry optimization, and electronic property calculations, while 1.0 × 10$^{-5}$ eV atom$^{-1}$ and 0.03 eV Å$^{-1}$ were applied for the energy and force converging to achieve high accuracy.

## Data availability

All relevant data that support the findings of this study are presented in the manuscript and supplementary information file. Source data are available from the corresponding author upon reasonable request.

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

## Acknowledgements

This work was financially supported by the National Natural Science Foundation of China (52070194, 51902347, 51822812, 21761132030), the Natural Science Foundation of Hunan Province (2020JJ5741), InnoSuisse through funding for the Swiss Competence Center for Energy Research (SCCER) Heat and Electricity Storage under contract number 1155-002545.

## Author contributions

X.F., X.O., W.Z. and Y.Y. conceived the project. X.O., W.Z. and Y.Y. supervised the project. X.F. synthesized the material and performed the battery tests with help from Y.L. and J.Z, while Y.Z.L. carried out the simulation. X.F., L.Z., L.S., B.Z. and G.H. carried out the material characterization. W.Z., X.O., L.Z. and Y.Y. helped to analyze data and propose mechanism. X.F., X.O., W.Z. prepared the manuscript with contributions from all other authors. W.Z., C.B. and Y.Y. contributed to discussion and revision of the manuscript.

## Competing interests

The authors declare no competing interests.
