## [Peer Review File · Nature Communications]

REVIEWER COMMENTS

Reviewer #1:

The reviewer enjoyed reading this work, it has all the required qualities from a top notch paper, the incentive is clear, the strategy is clear, and the results are promising.

The authors report the development of in-situ modification process to construct a uniform and conformal NASICON like protective layer and network on Ni rich NCM. The synthesized film interconnected the crystalline layered $\text{LiNi}_{0.88}\text{Co}_{0.09}\text{Mn}_{0.03}\text{O}_2$ (SC-NCM88) particles. The importance of the conductive thin ACEI is well explained and demonstrated throughout the paper. The reviewer has few minor comments on this work:

- 1) The authors should cite recent works in the field that demonstrated similar approach and results (e.g <https://pubs.acs.org/doi/abs/10.1021/acs.chemmater.7b00944> , <https://www.sciencedirect.com/science/article/abs/pii/S2405829720303251>, <https://pubs.acs.org/doi/10.1021/acsmi.0c10000>)
- 2) Since authors didn't measure directly the ionic conductivity of the $\text{Li}_{1.4}\text{Y}_{0.4}\text{Ti}_{1.6}(\text{PO}_4)_3$ (only comparing the cathode with and without coating) they should attempt to measure it, or tone down claims about its ionic conductivity

Reviewer #2:

The authors coated a single-crystalline NCM88 cathode with a fast-ion conductor (LYTP) and demonstrated that the coating improves the cycling stability of the single-crystal cathode at 4.4 V and an elevated temperature. Although this work does not represent a new approach, the result is encouraging that some of the disadvantages of single-crystal cathode Ni-rich layered cathodes can be resolved by a conformal coating. However, there are major flaws that need to be addressed.

1. In the method section, it is not clear what "a certain amount of mixture C" is. The authors should state that how much LYTP precursor was added to produce the final conformal coating layer. Was the LYTP mole fraction determined after the lithiation?
2. Although the authors claim that the coating layer is conformal, there is no concrete evidence that the secondary particle is uniformly coated by the LYTP layer. It is hard to believe that the LYTP precursor will remain conformal during the lithiation of the hydroxide precursor.
3. The thickness of the LYTP coating layer is roughly 20 nm in Fig. 1 e, but the LYTP layer is less than 5 nm after 200 cycles. Is the coating material chemically eroded during cycling. I am not even sure that the layer marked in Fig. 6h is LYTP. The marked layer could just be a damaged cathode surface. EDX analysis should be provided to confirm that the coating layer remains intact during and after cycling.
4. The electronic conductivity can be improved by the coating layer as the DFT calculation shows, but I am sure how a very thin coating layer can improve the bulk diffusion of Li ions. In addition, how the thin coating layer can reduce the c-axis contraction on the first charge. I am doubtful that the thin coating layer which is a hard ceramic material can constrain the contraction of the secondary particle. The dQ/dV curves for the coated and pristine cathodes are nearly identical in the first cycle, suggesting that the extent of the H2-H3 transition is similar for the two cathodes in the first cycle.
5. The interface between the LYTP layer and the cathode is critical to determining the mobility of Li ions and the chemical stability. Please provide a high-resolution TEM image of the interface, which should not be too difficult as the authors already have extensive TEM data. Is the interface sharp or is there a reaction layer?
6. Examining the TEM image in Fig. 6f, there is no coating material on the particle surface. If the coating layer is indeed 20 nm thick, it should be easily seen at this scale. Assuming that the image is a bright-field image, the dark contrast in the image could be due to structural defects. Fig. 6g also contains broken 003 lattice fringes, forming subgrain domains. Although there are no visible

cracks, the coated particle appears to be also highly strained and it is expected that cracks will eventually form even in the coated cathode. I believe that Fig. 6i is overstating the result.

7. Please remove Fig. 2 g. This figure is redundant and is misleading because the coating layer appears as a solid electrolyte. Fig. 1a is sufficient to show the coating layer.

Reviewer #3:

In this work, X. Fan and et al suggested the $\text{Li}_{1.4}\text{Y}_{0.4}\text{Ti}_{1.6}(\text{PO}_4)_3$ (LYTP) protective layers on single-particle Ni-rich NCM cathode for improving the high voltage cycling characteristic. The LYTP modification enables the Ni-rich cathode to attain an improved high voltage cycle life up to 85 % after 1000 cycles with 4.4 V, even in the pouch cell configuration with a commercially available active loading density. Using the X-ray diffraction analysis, DFT calculation, and precise surface compositional analysis, the authors tried to understand the origin of the improvement comprehensively. The electrochemical performance achieved here is very impressive. However, it is hard to rule out the possibility of miss-interpretation at some parts, and there are many similar cases in the coating and doping papers for Ni-rich cathode published so far. I suggest the authors address the following issues more precisely and publish their paper in a more specific journal.

1. Possibility of Ti doping - According to the previous report, Ti can be doped in the layered structure. The authors need to clarify why they ruled out the possibility of doping here even in synthesizing the NCM unusually high temperature 820 °C. Since the previous report suggesting a similar synthetic route claimed the Ti doping in layered oxide cathode. (Advanced Functional Materials, Volume 29, Issue 13 March 28, 2019 1808825, Simultaneously Dual Modification of Ni-Rich Layered Oxide Cathode for High-Energy Lithium-Ion Batteries) It is difficult to rule out the possibility of doping by SEM elemental mappings alone. A more accurate compositional analysis may be required here.

2. Suppression of H2-H3 phase transition - Authors claim that the 3D network would suppress the destructive H2-H3 phase transition. It has been generally shown that the phase transition of active material is kinetically suppressed or kinetically not achieved, but the authors suggested that the intrinsic thermodynamic phase transition will be limited in a better kinetic environment. Authors should provide supporting grounds or logic for this, or consider suggesting that a coherency exterior coating of the surface can limit the change in a lattice with citing a recent paper (Nature Materials volume 20, pages 84–92 (2021) Bulk fatigue induced by surface reconstruction in layered Ni-rich cathodes for Li-ion batteries)

3. In-operando XRD - When looking at the in-operando XRD data of LYTP@CS-NCM88 (Fig. 4 b), the peak positions of XRD have not changed since the discharge profile corresponding to approximately 20 hours. Moreover, in Fig 4d, the early XRD pattern of LYTP@CS-NCM88 still clearly shows the right shoulder peak from the monoclinic phase, while the control sample does not have this monoclinic phase, which indicates that control and coated samples do not match their state of charge in this figure. This observation often happens when the local area of the electrode being measured is not properly activated because of the contact or air bubble issues. I recommend that the author clarify this issue since this is critical to the author's main claim regarding the suppression of H2-H3 lattice change.

4. DFT calculation part

It will help readers read this paper by providing a more formal local atomic configuration used in DFT calculation. Besides, providing partial DOS for each element, if possible, will also promote a deeper understanding of this.

Dear Editor,

We appreciate all reviewer's for their positive and valuable suggestions to our manuscript. We have made all necessary changes suggested by the reviewers and marked them green in the revised manuscript (MS). Below are the point-to-point responses to editor's and reviewer's comments.

Reviewer 1

The reviewer enjoyed reading this work, it has all the required qualities from a top notch paper, the incentive is clear, the strategy is clear, and the results are promising.

The authors report the development of in-situ modification process to construct a uniform and conformal NASICON like protective layer and network on Ni rich NCM. The synthesized film interconnected the crystalline layered $\text{LiNi}_{0.88}\text{Co}_{0.09}\text{Mn}_{0.03}\text{O}_2$ (SC-NCM88) particles. The importance of the conductive thin CEI is well explained and demonstrated throughout the paper.

The reviewer has few minor comments on this work:

1) The authors should cite recent works in the field that demonstrated similar approach and results (e.g

<https://pubs.acs.org/doi/abs/10.1021/acs.chemmater.7b00944>,

<https://www.sciencedirect.com/science/article/abs/pii/S2405829720303251>,

<https://pubs.acs.org/doi/10.1021/acsami.0c10000>)

Reply: Thanks for the reviewer's valuable comment. We have cited the related literatures ^{R1-R3} as Reference 22-24 and added necessary description in page 4 in the revised manuscript, which are all marked in green (P4, line 23-25).

References

R1. Cheng J, Sivonxay E, Persson KA. Evaluation of amorphous oxide coatings for high-voltage Li-ion battery applications using a first-principles framework. *ACS Appl. Mater. Interfaces* **12**, 35748-35756 (2020).

R2. Rosy, *et al.* Alkylated $\text{Li}_x\text{Si}_y\text{O}_z$ coating for stabilization of Li-rich layered oxide cathodes. *Energy Storage Mater* **33**, 268-275 (2020).

R3. Kazyak E, *et al.* Atomic layer deposition of the solid electrolyte garnet $\text{Li}_7\text{La}_3\text{Zr}_2\text{O}_{12}$. *Chem. Mater.* **29**, 3785-3792 (2017).

2) Since authors didn't measure directly the ionic conductivity of the $\text{Li}_{1.4}\text{Y}_{0.4}\text{Ti}_{1.6}(\text{PO}_4)_3$ (only comparing the cathode with and without coating) they should attempt to measure it, or tone down

claims about its ionic conductivity.

Reply: Thanks for the reviewer's good suggestion. We prepared the pure $\text{Li}_{1.4}\text{Y}_{0.4}\text{Ti}_{1.6}(\text{PO}_4)_3$ at temperature of 820 °C and tested its ionic conductivity, as displayed in **Figure S11** and **Table S2**. The ionic conductivity of pure $\text{Li}_{1.4}\text{Y}_{0.4}\text{Ti}_{1.6}(\text{PO}_4)_3$ is $5.8 \times 10^{-4} \text{ S cm}^{-1}$ at 25 °C, which is higher than that of pristine SC-NCM88 ($6.31 \text{ E}^{-5} \text{ S cm}^{-1}$) and LYTP@SC-NCM88 ($9.49 \text{ E}^{-5} \text{ S cm}^{-1}$) as shown in **Figure 1h** and **Table S2**. The result illustrates that the classical NASICON-type $\text{Li}_{1.4}\text{Y}_{0.4}\text{Ti}_{1.6}(\text{PO}_4)_3$ modification facilitates to improve the conductivity of micron-sized SC-NCM88 particles.^{R4} All modified images and new table are shown below for easy review. The description of pure $\text{Li}_{1.4}\text{Y}_{0.4}\text{Ti}_{1.6}(\text{PO}_4)_3$ is added in the revised manuscript and marked in green (P8, line 15, 19-20).

Figure S11. Li-ion conductivity for pure $\text{Li}_{1.4}\text{Y}_{0.4}\text{Ti}_{1.6}(\text{PO}_4)_3$ at different temperatures

Figure 1h. STEM image of LYTP@SC-NCM88

Table S2. The ionic conductivities of pristine LYTP, pristine SC-NCM88, and 1% LYTP@SC-NCM88

Sample	Ionic conductivity (S/cm)		
	25°C	45°C	55°C
Pristine LYTP	5.91E-04	1.39E-03	1.98E-03
Pristine SC-NCM88	6.31E-05	1.38E-04	1.53E-04
1.0%LYTP@SC-NCM88	9.49E-05	1.56E-04	1.76E-04

References

R4. Rossbach A., Tietz F. & Grieshammer S. Structural and transport properties of lithium-conducting NASICON materials. *J Power Sources* **391**, 1-9 (2018).

Reviewer 2

The authors coated a single-crystalline NCM88 cathode with a fast-ion conductor (LYTP) and demonstrated that the coating improves the cycling stability of the single-crystal cathode at 4.4 V and an elevated temperature. Although this work does not represent a new approach, the result is encouraging that some of the disadvantages of single-crystal cathode Ni-rich layered cathodes can be resolved by a conformal coating. However, there are major flaws that need to be addressed.

1. In the method section, it is not clear what “a certain amount of mixture C” is. The authors should state that how much LYTP precursor was added to produce the final conformal coating layer. Was the LYTP mole fraction determined after the lithiation?

Reply: Thanks for your helpful comments. The description of “a certain amount of mixture C” is inaccurate, which is corrected as “After that, the $\text{Ni}_{0.88}\text{Co}_{0.09}\text{Mn}_{0.03}(\text{OH})_2$ was added into the aforementioned solution with mild stirring”. Meanwhile, following the reviewer’s suggestion, we have added the detailed preparation of 1%LYTP@SC-NCM88 in the Methods section of revised manuscript. "Specifically, to synthesize the of 1% LYTP@SC-NCM88 sample, 0.47g LiNO_3 , 0.75g $\text{Y}(\text{NO}_3)_3 \cdot 6\text{H}_2\text{O}$, 1.44g H_3PO_4 (the molar ratio of Li : Y : P = 1.4 : 0.4 : 3), and a stoichiometric ratio of 2.68g $\text{Ti}(\text{C}_4\text{H}_9\text{O})_4$ were dissolved and mixed homogeneously in ethyl alcohol solutions. After that, 187.9g $\text{Ni}_{0.88}\text{Co}_{0.09}\text{Mn}_{0.03}(\text{OH})_2$ was added into the aforementioned solution with mild stirring. The LYTP-precursor@ $\text{Ni}_{0.88}\text{Co}_{0.09}\text{Mn}_{0.03}(\text{OH})_2$ particles were finally obtained through washing, and vacuum drying at 110 °C overnight. To get LYTP@SC-NCM88, this LYTP-precursor@ $\text{Ni}_{0.88}\text{Co}_{0.09}\text{Mn}_{0.03}(\text{OH})_2$ was thoroughly mixed with 90g $\text{LiOH} \cdot \text{H}_2\text{O}$ (Li: M ratio = 1.06:1) and calcined at 820°C for 10 h in oxygen atmosphere. The material was finally ground by air-jet mill and was stored in vacuum package. "After calcination, the LYTP mole fraction in the LYTP@SC-NCM88 composite was determined according to the rechecked ICP results (**Table S1**). For

1.0%LYTP@SC-NCM88 sample, the mole ratio of LYTP: SC-NCM88 is 0.007:1, which is transformed into the mass ratio of 0.01:1. The Table S1 is copied below for easy review and the detailed description was marked in green in the manuscript (P23, line 17-25; P24, line 1-2).

Table S1. Chemical compositions of the pristine SC-NCM88 and modified LYTP@SC-NCM88 cathodes measured by inductively coupled plasma analysis.

Sample	ICP (wt%)					
	Ni	Co	Mn	Ti	Y	P
Pristine SC-NCM88	52.831	5.498	1.802	/	/	/
0.5%LYTP@SC-NCM88	52.714	5.527	1.568	0.092	0.043	0.112
1.0%LYTP@SC-NCM88	52.271	5.260	1.839	0.196	0.092	0.238
1.5%LYTP@SC-NCM88	51.770	5.591	1.608	0.282	0.129	0.339
Sample	Measured molar ration					
	Ni	Co	Mn	Ti	Y	P
Pristine SC-NCM88	0.877	0.091	0.032	/	/	/
0.5%LYTP@SC-NCM88	0.881	0.092	0.028	0.0019212	0.0004879	0.0036238
1.0%LYTP@SC-NCM88	0.878	0.088	0.033	0.0040852	0.0010304	0.007699
1.5%LYTP@SC-NCM88	0.874	0.094	0.029	0.0058903	0.0014535	0.0109607

2. Although the authors claim that the coating layer is conformal, there is no concrete evidence that the secondary particle is uniformly coated by the LYTP layer. It is hard to believe that the LYTP precursor will remain conformal during the lithiation of the hydroxide precursor.

Reply: Thanks for your helpful comments. In this work, the EDS mapping of the cross-sectional 1% LYTP@SC-NCM88 indicates that the Y and Ti elements conformally anchor on the surface of the particles (**Figure S7c**). In order to further confirm the uniform modification of LYTP on the surface of SC-NCM88, the cross sectional electron probe micro-analyzer (EPMA) was applied to analyze the distributions of chemical compositions (**Figure 1d** and **Figure S8**). It is clear that Ti, P elements mainly wraps on the surface of particles while Ni, Co, and Mn elements enriches in the core of the particles, confirming the uniform distribution of LYTP layer due to the formation of quasi core-shell-structured LYTP@SC-NCM88. Moreover, the gaps of inter-particles are filled with LYTP, demonstrating the 3D

interconnected LYTP network, which is able to facilitate the Li⁺-ion transport among the dispersed micron-sized SC-NCM particles. We have added the description of EPMA result and marked in green in the revised manuscript (P7, line 12-19), and all related images are copied below for easy review.

Figure 1d. The cross-section EPMA image of 1% LYTP@SC-NCM88 with the corresponding selected area LYTP mapping results of Ni, Co, Mn, Ti, and P elements.

Figure S7c. Elemental SEM mappings of Ni, Co, Mn, Y, and Ti images of 1% LYTP@SC-NCM88.

Figure S8. The cross-sectional EPMA image of 1% LYTP@SC-NCM88 with the corresponding selected area LYTP mapping results of Ni, Co, Mn, Ti, and Y element.

3. The thickness of the LYTP coating layer is roughly 20 nm in Fig. 1e, but the LYTP layer is less than 5 nm after 200 cycles. Is the coating material chemically eroded during cycling? I am not even sure that the layer marked in Fig. 6h is LYTP. The marked layer could just be a damaged cathode surface. EDX analysis should be provided to confirm that the coating layer remains intact during and after cycling.

Reply: Thanks for your helpful comments. We have re-measured the 1% LYTP@SC-NCM88 sample after 200 cycles by TEM characterization (**Figure 6b** and **Figure S32**). Our observations show that the thickness of LYTP layer varies a bit at different regions. To get more accurate data of the thickness evolution upon cycling, we have checked more regions of different cycled particles, our captured data indicate that the thickness of LYTP layers range from 5 nm to 20 nm. Therefore, we put an image with the thickness of 17.5 nm for demonstration. We have added the relevant description in the text. In order to further confirm if the LYTP modification layer maintains intact after cycling, the STEM elemental mappings analysis was employed to check the Ti element distribution of 1% LYTP@SC-NCM88 sample after 200 cycles. As displayed in **Figure S31**, the representative Ti element uniformly anchors on the surface of SC-NCM88, demonstrating that the LYTP modification layer keeps chemical stability and structural integrity during the long-term cycling. Base on above results, we can safely claim that LYTP modification layer holds a thickness of 5-20nm and maintain chemical stability and robust structure after the long-term cycling. We have replaced **Figure 6b** and **Figure S31-S32** as well as added the related description of TEM result after cycling in the revised manuscript (P19, line 15-25).

Figure 6b. Post-mortem HRTEM and magnified HRTEM at selected area images for (b, b1, b2) 1% LYTP@SC-NCM88 after 200 cycles.

Figure S31. STEM elemental mappings of Ni, Ti, and O for 1% LYTP@SC-NCM88 after 200 cycles.

Figure S32. (a) TEM and (b) HRTEM images of pristine SC-NCM88. (c) TEM and (d-f) HRTEM image of surface region for 1% LYTP@ SC-NCM88 after 200 cycles.

4. The electronic conductivity can be improved by the coating layer as the DFT calculation shows, but I am sure how a very thin coating layer can improve the bulk diffusion of Li ions. In addition, how the thin coating layer can reduce the c-axis contraction on the first charge. I am doubtful that the thin coating layer which is a hard ceramic material can constrain the contraction of the secondary particle. The dQ/dV curves for the coated and pristine cathodes are nearly identical in the first cycle, suggesting that the extent of the H2-H3 transition is similar for the two cathodes in the first cycle.

Reply: Thanks for your valuable comments. We agree with reviewer's opinion that it is difficult to confirm how a coating layer improves the bulk diffusion of Li⁺ through DFT calculation and other techniques. In our case, the established 3D network of LYTP modification facilitates the Li⁺ transfer between the interconnected SC-NCM88 particles as the LYTP has a NASICON-type structure with high Li⁺ conductivity. Moreover, it is worth noting that the Li⁺ goes smoothly through the well-maintained electrode/electrolyte interface due to the 3D conformal protection of LYTP modification and robust Ti-O lattice surface doping, which is confirmed by the decreased charge transfer resistance (R_{ct}) of LYTP@SC-NCM88 electrode after 200 cycles (**Figure S29**). We changed the related description in the DFT calculation section (P16, line 13-14).

We agree that the LYTP coating is a hard ceramic material, however, it maintains firmly conformal and intact on the surface of SC-NCM88 even after 200 cycles (**Figure 6b**). Based on the *in-situ* XRD analysis, the significant change of lattice parameter of c-axis presents an expansion/contraction of the interlayer spacing, coupled with the H2-H3 phase transition. At highly delithiated state, it is easy to induce oxygen loss and structural mismatch between layered phase and spinel/rock-salt phase, which would result in the severe microstrain and even the fatigued phase in the bulk structure along c-axis.^{R5, R6} However, the ultra-conformal LYTP modification effectively reduced the degree of lattice mismatch at interphase, which is also able to suppress the c-lattice parameter contraction and to improve the reversibility of H2-H3 phase transition.^{R5-R7} Additionally, the trace Ti-doping at the surface/subsurface (**Figure S10**) would be helpful to suppress the formation of undesired surface

spinel/rock-salt phases and to prevent the oxygen loss, maintaining surface structural stability and bulk structural integrity.^{R8} This phenomenon is also confirmed by the second cycle of *in-situ* XRD patterns for pristine SC-NCM88 and 1% LYTP@SC-NCM88 (**Figure S21**).

In order to distinguish the difference of H2-H3 transition for pristine SC-NCM88 and 1% LYTP@SC-NCM88, the initial cycle of cyclic voltammetry (CV) curves are displayed in **Figure S18**. The H2-H3 transition corresponds to the peak at ~4.15 V, induced by the lattice contraction in the *c*-axis. The peak intensity at ~4.15 V of 1% LYTP@SC-NCM88 is weaker than that of pristine SC-NCM88, illustrating the smaller anisotropic lattice volume variation after LYTP modification. Therefore, both of the dQ/dV is similar but different, as the peak intensity of 1% LYTP@SC-NCM88 is weaker than the pristine SC-NCM88 due to the alleviation of H3 phase transformation.^{R9} The related descriptions of **Figure S10** (P8, line 5-10), **Figure S18** (P12, line 18-23) and **Figure S21** (P14, line 18-21; P15, line 22-25, P16, line 1) were marked in green in the manuscript and copied below for easy review.

References

- R5. Xu G.-L. *et al.* Building ultraconformal protective layers on both secondary and primary particles of layered lithium transition metal oxide cathodes. *Nat. Energy* **4**, 484-494 (2019).
- R6. Xu C, *et al.* Bulk fatigue induced by surface reconstruction in layered Ni-rich cathodes for Li-ion batteries. *Nat. Mater.* **20**, 84-92 (2021).
- R7. Xu C, Reeves PJ, Jacquet Q, Grey CP. Phase behavior during electrochemical cycling of Ni-rich cathode materials for Li-ion batteries. *Adv Energy Mater* **11**, 2003404 (2021).
- R8. Yang H. *et al.* Simultaneously dual modification of Ni-rich layered oxide cathode for high-energy lithium-ion batteries. *Adv. Funct. Mater.* **29**, 1808825 (2019).
- R9. Ryu H-H, Park K-J, Yoon CS, Sun Y-K. Capacity fading of Ni-rich Li[Ni_xCo_yMn_{1-x-y}]O₂ (0.6 ≤ x ≤ 0.95) cathodes for high-energy-density lithium-ion batteries: bulk or surface degradation? *Chem Mater* **30**, 1155-1163 (2018).

Figure S10. Quantification of Ni, Co, Mn, O, Ti, Y, P resulting from STEM-EDS mapping for 1%LYTP@SC-NCM88 cathode material.

Figure S18. The overlapped and individual CV curves for (a, b) pristine SC-NCM88 and (a, c) 1% LYTP@SC-NCM88.

Figure S21. In-situ XRD stacked patterns for pristine SC-NCM88 (a) and 1% LYTP@SC-NCM88 cathodes (b) during the first 2 cycle's charge-discharge process.

5. The interface between the LYTP layer and the cathode is critical to determining the mobility of Li^+ ions and the chemical stability. Please provide a high-resolution TEM image of the interface, which should not be too difficult as the authors already have extensive TEM data. Is the interface sharp or is there a reaction layer?

Reply: Thanks for the helpful suggestions. We have re-checked and replaced the TEM data, the new TEM images displayed in **Figure 1e-f** and **Figure S7** in the revised manuscript. The interface between LYTP layer and cathode of the 1% LYTP@SC-NCM88 is revealed by TEM image (**Figure 1e**), which exhibits a bulk/coating configuration with regular reaction layer. The interplanar spacings of 0.47 nm from the bulk phase is indexed to the (003) plane of SC-NCM88. The LYTP modification skin with a thickness of ≈ 20 nm is consisted with stacked nanograins, which is clearly indexed into the (113) planes of LYTP (**Figure 1f**). The results demonstrate that the in-situ modification guarantees the LYTP thin layer anchoring around the grain boundaries of SC-NCM88, naturally constructing a 3D conductive network by integrating the intrinsic fast Li^+ transport of LYTP. The related images are copied below for easy review and the revision is marked in green in the manuscript (P7, line 22-25).

Figure 1e-f. (e) TEM and (f) HRTEM of 1% LYTP@SC-NCM88 images.

Figure S7. (a) TEM, (b) HRTEM and (c) elemental SEM mappings of Ni, Co, Mn, Y, and Ti images of 1% LYTP@SC-NCM88.

6. Examining the TEM image in Fig. 6f, there is no coating material on the particle surface. If the coating layer is indeed 20 nm thick, it should be easily seen at this scale. Assuming that the image is a bright-field image, the dark contrast in the image could be due to structural defects. Fig. 6g also contains broken 003 lattice fringes, forming subgrain domains. Although there are no visible cracks, the coated particle appears to be also highly strained and it is expected that cracks will eventually form even in the coated cathode. I believe that Fig. 6i is overstating the result.

Reply: Thanks for your helpful suggestions. We have re-measured the 1% LYTP@SC-NCM88 electrode after 200 cycles and replaced the TEM images in **Figure 6b** and **Figure S32c-f**. As confirmed by the HRTEM image in **Figure 6b**, a uniform and compact coating layer is firmly adhered to the primary particle, and region b1 and region b2 are indexed into the LYTP with a space group of R-3c and layered structure of NCM with a space group of $R\bar{3}m$, respectively. The coating thickness of LYTP

is about 17.5 nm, which is consistent with the thickness (5-20 nm) of uncycled 1% LYTP@SC-NCM88 electrode.

The dark contrast in a bright-field image easily makes us confusion about the integrity of layered structure, which would be attributed to the various thickness of SC-NCM88 electrode during the FIB treatment or structural defects because of harsh condition operation. To clearly clarify the point, we replaced the HAADF-STEM image of cycled 1% LYTP@SC-NCM88 electrode, as displayed in **Figure 6i-6j**. Although the surface of SC-NCM88 shows slight disordered crystalline due to the corrosion of electrolyte, its structure keeps clear layered lattices from the interior bulk (**Figure 6j**) to outermost surface region (**Figure 6i**). The above results confirm that the LYTP layer is electrochemically and structurally stable even at high voltages operation and is able to mitigate a cascade of parasitic interface reactions between SC-NCM88 and electrolyte, achieving an enhanced interphase stability of 1% LYTP@SC-NCM88 even in the long-term cycling. Honestly, the harsh testing condition inevitably exerts a slight influence on the structure stability of electrode, so the randomly disordered structure distribution is unavoidable in the small region of cycled LYTP@SC-NCM88 electrode.^{R10} Therefore, we changed the ultra-stability to enhanced-stability in the structural schematic illustration of **Figure 6k** and modified the related description in the revised manuscript (P19, line 17-19; P20, line 17-21). The modified images are copied below for easy review.

References

R10. Han B, *et al.* Enhancing the structural stability of Ni-rich layered oxide cathodes with a preformed Zr-concentrated defective nanolayer. *ACS Appl Mater Interfaces* **10**, 39599-39607 (2018).

Figure S32c-f. (c) TEM and (d-f) HRTEM image of surface region for 1% LYTP@ SC-NCM88 after 200 cycles.

Figure 6. Intraparticle structural evolution after long-term cycling. Post-mortem HRTEM and magnified HRTEM at selected area images for (b, b1, b2) 1% LYTP@SC-NCM88 after 200 cycles. Low-magnification HAADF-STEM image of FIB-cross section for the magnified HAADF-STEM images taken from the corresponding surface areas for (h-j) 1%LYTP@SC-NCM88 after 200 cycles. (k) Schematic illustration of the structure evolution and internal crack difference for pristine SC-NCM88 and 1% LYTP@SC-NCM88 particles during prolonged cycling.

7. Please remove Fig.2 g. This figure is redundant and is misleading because the coating layer appears as a solid electrolyte. Fig. 1a is sufficient to show the coating layer.

Reply: Thanks for your helpful suggestion. We agree with reviewer's opinion and the Figure 2g was removed in the revised manuscript.

Reviewer 3

In this work, X. Fan and et al suggested the $\text{Li}_{1.4}\text{Y}_{0.4}\text{Ti}_{1.6}(\text{PO}_4)_3$ (LYTP) protective layers on single-particle Ni-rich NCM cathode for improving the high voltage cycling characteristic. The LYTP modification enables the Ni-rich cathode to attain an improved high voltage cycle life up to 85 % after 1000 cycles with 4.4 V, even in the pouch cell configuration with a commercially available active loading density. Using the X-ray diffraction analysis, DFT calculation, and precise surface compositional analysis, the authors tried to understand the origin of the improvement comprehensively. The electrochemical performance achieved here is very impressive. However, it is hard to rule out the possibility of miss-interpretation at some parts, and there are many similar cases in the coating and doping papers for Ni-rich cathode published so far. I suggest the authors address the following issues more precisely and publish their paper in a more specific journal.

1. Possibility of Ti doping - According to the previous report, Ti can be doped in the layered structure. The authors need to clarify why they ruled out the possibility of doping here even in synthesizing the NCM unusually high temperature 820 °C. Since the previous report suggesting a similar synthetic route claimed the Ti doping in layered oxide cathode. (Advanced Functional Materials, Volume29, Issue 13 March 28, 2019 1808825, Simultaneously Dual Modification of Ni-Rich Layered Oxide Cathode for High-Energy Lithium-Ion Batteries) It is difficult to rule out the possibility of doping by SEM elemental mappings alone. A more accurate compositional analysis may be required here.

Reply: Thanks for the valuable comments. In our work, as we mainly focus on the coating effort of LYTP on the surface of SC-NCM88, we did not confirm if the Ti doping is exist in the previous edition. Herein, we employ STEM-EDS line scanning to confirm the foreign elements distribution of Ti, Y and even try to analyze the accurate composition in the bulk structure of SC-NCM88. As displayed in **Figure S10**, the Ti depth profile demonstrates that the Ti-based coating layer and gradient doping of Ti in the subsurface regions came true simultaneously. Meanwhile, it is clearly shown that the contents of Ni, Co, and Mn are barely changed from the subsurface to interior bulk. Therefore,

these distinctions indicate that the LYTP modification skin with a thickness of 5-20 nm well wraps on the primary SC-NCM88 particle, accompanying with trace doping of Ti in the subsurface of SC-NCM88. This phenomenon is in consistent with previous reports.^{R8, R11} The simultaneous LYTP coating and trace Ti-doping exert synergistic modification effect on the phase transformation and structure stability of SC-NCM88. Following the reviewer's suggestion, we have added the description of trace Ti-doping at subsurface region and marked in green in the revised manuscript (P5, 8-10; P8, line 5-10). The **Figure S10** is copied below for easy review.

References

R8. Yang H. *et al.* Simultaneously dual modification of Ni-rich layered oxide cathode for high-energy lithium-ion batteries. *Adv. Funct. Mater.* **29**, 1808825 (2019).

R11. Li W, *et al.* Regulating the grain orientation and surface structure of primary particles through tungsten modification to comprehensively enhance the performance of Nickel-rich cathode materials. *ACS Appl Mater Interfaces*, **12**, 47513-47525 (2020).

Figure S10. TEM images and EDS elemental mapping of Ni, Co, Mn, O, Ti, Y, P for 1%LYTP@SC-NCM88 cathode material.

2. Suppression of H2-H3 phase transition - Authors claim that the 3D network would suppress the destructive H2-H3 phase transition. It has been generally shown that the phase transition of active material is kinetically suppressed or kinetically not achieved, but the authors suggested that the intrinsic thermodynamic phase transition will be limited in a better kinetic environment. Authors should provide supporting grounds or logic for this, or consider suggesting that a coherency exterior coating of the surface can limit the change in a lattice with citing a recent paper (Nature Materials volume 20, pages 84-92 (2021) Bulk fatigue induced by surface reconstruction in layered Ni-rich

cathodes for Li-ion batteries).

Reply: Thanks for your helpful suggestion. We would like to clarify our logic more clear. The 3D network structure in 1%LYTP@SC-NCM88 plays a positive role to enhance the conductivity of electron and Li⁺-ion by interconnecting the modified particles, while the surface modification suppresses the destructive H2-H3 phase transition of SC-NCM88. According to the recent report, the capacity degradation derives from the lattice mismatch with high interfacial lattice strain between the well-ordered layered structure and spinel/rock-salt reconstructed surface, especially at the high charging state over 4.2 V, which is accompanied with H2-H3 phase transition.^{R5, R6} During the long-term cycling, the surface reconstruction layers will exert pin effect on the adjacent bulk layered structure, gradually leads to a fatigued phase generation, which is detrimental to the Li⁺ transfer smoothly and to the reversibility of capacity.

In our work, the conformal LYTP skin is able to prevent the bulk fatigue of SC-NCM88 as the alleviation of cathode/electrolyte interface side reaction. When tested at harsh condition of high cut-off voltage (>4.2 V) and elevated temperature for 1% LYTP@SC-NCM88 electrode, the strong Ti–O bond at surface is helpful to suppress the formation of undesired surface spinel/rock-salt phases during the cycling. Substantially, the LYTP modification effectively reduces the lattice mismatch (**Figure 6j** and **Figure S32e-f**) and decreases the fraction of fatigued phase within aged electrode, finally suppressing the lattice contraction and boost the reversibility of H2-H3 phase transition.^{R6, R7} Thus, the LYTP modification alleviates the pinning effect of reconstructed lattice mismatch, efficiently enhancing the phase transition reversibility and structural stability. At the other hand, the 3D conductive network of LYTP improves the conductivity of Li⁺ ions and electron in LYTP@SC-NCM88 cathode through interconnecting the particles. Following the reviewer's suggestion, we have added the necessary references and the description of LYTP modification on intrinsic thermodynamic phase transition for aged SC-NCM, as displayed in the Reference section in the revised manuscript, respectively (P20, line 17-21; P22, line 11-14; P23, line 3-7). The **Figure 6j** and **Figure 32e-f** are

copied below for easy review.

References:

- R5. Xu G.-L. *et al.* Building ultraconformal protective layers on both secondary and primary particles of layered lithium transition metal oxide cathodes. *Nat. Energy* 4, 484-494 (2019).
- R6. Xu C, *et al.* Bulk fatigue induced by surface reconstruction in layered Ni-rich cathodes for Li-ion batteries. *Nat. Mater.* 20, 84-92 (2021).
- R7. Xu C, Reeves PJ, Jacquet Q, Grey CP. Phase behavior during electrochemical cycling of Ni-rich cathode materials for Li-ion batteries. *Adv Energy Mater* 11, 2003404 (2021).

Figure 6j. Low-magnification HAADF-STEM image of FIB-cross section for the magnified HAADF-STEM images taken from the corresponding surface areas (j) 1%LYTP@SC-NCM88 after 200 cycles.

Figure S32. (e-f) HRTEM image of surface region for 1% LYTP@ SC-NCM88 after 200 cycles.

3. In-operando XRD - When looking at the in-operando XRD data of LYTP@SC-NCM88 (Fig. 4 b), the peak positions of XRD have not changed since the discharge profile corresponding to approximately 20 hours. Moreover, in Fig 4d, the early XRD pattern of LYTP@SC-NCM88 still clearly shows the right shoulder peak from the monoclinic phase, while the control sample does not have this monoclinic phase, which indicates that control and coated samples do not match their state of charge in this figure. This observation often happens when the local area of the electrode being measured is not properly activated because of the contact or air bubble issues. I recommend that the author clarify this issue since this is critical to the author's main claim regarding the suppression of H2-H3 lattice change.

Reply: Thanks for the valuable comments. After we rechecked the raw data of LYTP@SC-NCM88, the descriptions of **Figure 4b** and **Figure 4d** may be inaccurate due to the interval suspension during the *in-situ* XRD test. As shown in **Figure R1a**, there are ~3.5h rest during the first discharge cycle, leading to the mismatch pattern in **Figure 4b** (unchanged profile at the fully discharge state). Actually, the *in-situ* XRD pattern of LYTP@SC-NCM88 after removal of suspension period (**Figure R1b**) is in consistent with pristine SC-NCM88 (**Figure 4a**). For better understanding, we have modified the **Figure 4b**, as displayed in the revised manuscript.

Furthermore, the shoulder peak at 18.5° in **Figure 4d** is attributed to the monoclinic phase, which is detected at the charge state of 3.9 V, while the same peak presents at the charge state of 4.1V in **Figure 4c**, resulting in the inconsistent variation pattern between SC-NCM88 and 1%LYTP@SC-NCM88 samples. In order to reconfirm the mismatch of *in-situ* XRD pattern, we eliminated all potential issues of inappropriate test process (contact, air bubble ect. al.) and re-measured the *in-situ* XRD variation of LYTP@SC-NCM88. The reasonable phase evolution of LYTP@SC-NCM88 is got and is shown in **Figure S22** and **Figure 4d**, which is consistent with the previous patterns of LYTP@SC-NCM88 in revised **Figure 4b**. Following the reviewer's valuable suggestion, we have changed the description of Figure 4 in the revised manuscript (P15, line 17-25; P16, line 1). The related

images are copied below for easy review.

Figure R1. *In-situ* XRD stacked patterns for (a) origin data and (b) amended results for 1% LYTP@SC-NCM88 cathodes during the first two cycles.

Figure 4a-e. Operando XRD characterization. Operando XRD characterization of the full contour plots and selected line patterns for (a, c) SC-NCM88 and (b, d) 1% LYTP@SC-NCM88 cathodes during the initial cycle in the voltage range of 2.75–4.6 V. (e) The variation of the c-axis parameter during the charging.

4. DFT calculation part

It will help readers read this paper by providing a more formal local atomic configuration used in DFT calculation. Besides, providing partial DOS for each element, if possible, will also promote a deeper understanding of this.

Reply: Thanks for your helpful advice. Following the reviewer's suggestion, we have further conducted DFT calculation by providing a more formal local atomic configuration. Four different types of relaxed crystal structures are chosen, while the 3D-difference charge density are depicted in **Figure 4h** and **Figure S25**. To simplify the calculation, LiNiO₂ structure is applied as the pristine model of SC-NCM88 (**Figure S25a**). Next, trace Ti-atoms are introduced into the surface site around the selected oxygen atoms for SC-NCM88-Ti (**Figure S25b**). Then, LYTP nanolayer is introduced to the surface with intimate bonding for SC-NCM88-LYTP (**Figure S25c**). Furthermore, the LYTP coating and trace Ti-doping are introduced simultaneously, as displayed in **Figure 4h**. Meanwhile, the corresponding 2D-difference charge density of four types are depicted in **Figure 4i-4j** and **Figure S26**. In comparison, there is an obvious electron transfer phenomenon between Ni, Co, Ti and the surrounding O atoms, which becomes more obvious with the introduction of surface Ti-doping and LYTP coating. Additionally, the degree of electron accumulation for Ti-O is higher than that of Ni-O and Co-O, which indicates that Ti-O has a stronger chemical bond, thus stabilizing the lattice oxygen during lithiation/delithiation.

Based on the difference charge density of atomic configuration, the Bader charge transfer of four types is also calculated, as displayed in **Figure 4k, 4l** and **Figure S27**. Through Bader charge transfer analysis, it is also seen that Ti atom transfers more electrons than Ni and Co atoms, which can enhance the surface oxygen stability. This result is consistent with the previous differential charge density results. During the long-term cycling with elevated temperature and high cut-off voltage, the improved surface stability will mitigate the oxygen release. Therefore, it can suppress the unwanted transformations from layered structure to disordered spinel/rock-salt phases, thus ensuring the superior

electrochemical and thermal performance.

Additionally, total and partial DOS for each element are also calculated by DFT calculation, as shown in **Figure 4f-g** in the revised manuscript. In addition, DOS analysis confirm that the introduction of surface Ti-doping and LYTP coating will induce the left shift of overall DOS and increase the electron concentration, indicating the enhancement of its conductivity. Following the reviewer's suggestion, we have added the DFT results in the revised manuscript (P16, 7-25; P17, line 1-4) and copied all related images below for easy review.

References:

R11. Li W, *et al.* Regulating the grain orientation and surface structure of primary particles through tungsten modification to comprehensively enhance the performance of Nickel-rich cathode materials. *ACS Appl Mater Interfaces*, **12**, 47513-47525 (2020).

Figure 4f-l. Density functional theory calculation. The total and partial density of states plots for (f) pristine SC-NCM88 and (g) 1% LYTP@SC-NCM88. (h) Relaxed crystal structures of 1% LYTP@SC-NCM88. (i) The corresponding 2D charge difference scheme and Bader charge transfer for (i, k) pristine SC-NCM88 and (j, l) 1% LYTP@SC-NCM88.

Figure S25. Relaxed crystal structures for (a) pristine SC-NCM88, (b) surface Ti-atoms doping (SC-NCM88-Ti), and (c) LYTP nanolayer coating (SC-NCM88-LYTP).

Figure S26. The corresponding 2D charge difference scheme for (a) surface Ti-atoms doping (SC-NCM88-Ti) and (b) LYTP nanolayer coating (SC-NCM88-LYTP).

Figure S27. The Bader charge transfer for (a) surface Ti-atoms doping (SC-NCM88-Ti) and (b) LYTP nanolayer coating (SC-NCM88-LiYTP).

REVIEWERS' COMMENTS

Reviewer #2 (Remarks to the Author):

The questions and comments I raised have been adequately addressed in the revised manuscript.

Reviewer #3 (Remarks to the Author):

I sincerely appreciate the author's enthusiastic response. The quality of this version of the manuscript has been much improved, and most of unclear part has been addressed well. As I pointed out in the first review, the electrochemical performance of this work is outstanding, even using the commercially available loading density and pouch cell configuration. However, it is difficult to give a high evaluation of the innovativeness or novelty of this study. Many coating papers and similar approaches have already been tried in this field, and the reasons for the increase in electrochemical properties described here have already been found in previous studies. Moreover, it is difficult to judge novelty to a paper simply by having good electrochemical properties. Although this study has high completeness and comparable data structure, I am not sure whether it should be published in a leading journal such as Nature Communications.

Referee 3#

I sincerely appreciate the author's enthusiastic response. The quality of this version of the manuscript has been much improved, and most of unclear part has been addressed well. As I pointed out in the first review, the electrochemical performance of this work is outstanding, even using the commercially available loading density and pouch cell configuration. However, it is difficult to give a high evaluation of the innovativeness or novelty of this study. Many coating papers and similar approaches have already been tried in this field, and the reasons for the increase in electrochemical properties described here have already been found in previous studies. Moreover, it is difficult to judge novelty to a paper simply by having good electrochemical properties. Although this study has high completeness and comparable data structure, I am not sure whether it should be published in a leading journal such as Nature Communications.

Response: Thanks for your comments. As far as we know, it is still challenging to synthesize single crystalline NCM materials with high Ni content ($\text{Ni} > 0.85$) in good quality and achieve good cycling capability at high operating voltage ($\geq 4.4\text{V}$ vs Li/Li^+) or high cycling rate ($> 2\text{C}$, $1\text{C} = 200\text{ mA g}^{-1}$), owing to the unavoidable parasitic reaction at cathode/electrolyte interphases and slow Li^+ transport inside of the microsized-particles. Therefore, the main novelty of the manuscript is developing an in-situ strategy to interconnect the single-crystalline $\text{LiNi}_{0.88}\text{Co}_{0.09}\text{Mn}_{0.03}\text{O}_2$ (SC-NCM88) particles by establishing a conductive $\text{Li}_{1.4}\text{Y}_{0.4}\text{Ti}_{1.6}\text{PO}_4$ (LYTP) network to enable faster Li^+ transport of the NCM particles and to improve its surface stability. In addition, we not only show the excellent performance of the modified SC-NCM88, but also propose the working mechanism of LYTP network by using different techniques. These new findings and newly achieved performance of single crystalline NCM materials will stimulate wide investigations and applications of Ni-rich NCM materials in single-crystalline states in the battery community. We believe that this pioneering work will be interesting for the broad readership of Nature Communications.